# Influence of the choice of insolation forcing on the results of a conceptual glacial cycle model

Gaëlle Leloup[1,2] and Didier Paillard[2]

[1]Agence Nationale pour la gestion des déchets radioactifs (ANDRA), 1 Rue Jean Monnet, 92290 Châtenay-Malabry, France
[2]Laboratoire des Sciences du Climat et de l'Environnement, LSCE/IPSL, CEA-CNRS-UVSQ-Université Paris-Saclay, 91198 Gif-sur-Yvette, France

**Correspondence:** Gaëlle Leloup (gaelle.leloup@lsce.ipsl.fr)

**Abstract.**

Over the Quaternary, the ice volume variations are "paced" by the astronomy. However, the precise way in which the astronomical parameters influence the glacial-interglacial cycles is not clear. The origin of the 100 kyr cycles over the last million year and of the switch from 40 kyr to 100 kyr cycles over the Mid Pleistocene Transition remain largely unexplained. By representing the climate system as oscillating between two states, glaciation and deglaciation, switching once a glaciation and a deglaciation thresholds are crossed, the main features of the ice volume record can be reproduced (Parrenin and Paillard, 2012). However, previous studies have only focused on the use of a single summer insolation as input. Here, we use a simple conceptual model to test and discuss the influence of the use of different summer insolation forcings, having different contributions from precession and obliquity, on the model results. We show that some features are robust. Specifically, to be able to reproduce the frequency shift over the Mid Pleistocene Transition, while having all other model parameters fixed, the deglaciation threshold needs to increase over time, independently of the summer insolation used as input. The quality of the model data agreement however depends on the chosen type of summer insolation and time period considered.

## 1 Introduction

Paleoclimate records over the Quaternary (last 2.6 Myr), such as ice cores (Jouzel et al., 2007) or marine cores (Lisiecki and Raymo, 2005) show a succession of oscillations. These oscillations are due to the build up and retreat of northern continental ice sheets, corresponding to respectively cold and warm periods known as the glacial - interglacial cycles. Over the last million year, there is an alternance of long glaciations, followed by quick deglaciations leading to cycles of a period of 100 kyr. Yet, on the earliest part of the record, glacial-interglacial cycles are mostly dominated by a frequency of 41 kyr with lower amplitude. Spectral analysis of the record (Hays et al., 1976) have revealed that astronomical frequencies are imprinted into the ice volume record, suggesting a strong link between insolation and the glacial - interglacial cycles.

The nature and physics of this link has been a central question since the discovery of previous warm and cold periods, and long before the obtention of continuous $\delta^{18}O$ records. It is known that the variations of annual total energy are of too low amplitude to explain such changes (Croll, 1864). Therefore, focus has been set on seasonal variations. Croll (1864) assumed that glaciations were linked to colder winter. In contrast, the idea that the decisive element for glaciation was the presence of

cold summers, due to reduced summer insolation, at latitudes of the Northern Hemisphere critical for ice sheet growth (65° N)
was taken up by Milankovitch. He made it the key element of his ice age theory (Berger, 2021). For conceptual models, this
raises the question of which insolation to use as input. When summer insolation is used, this questions the definition of summer :
should it be defined as a specific single day, like the summer solstice; the astronomical summer between the two equinoxes; or a
fixed number of days around the solstice. This choice leads to very different forcings with different contributions from obliquity
and precession. For ESMs and climate models, insolation is computed at each timestep for each grid area, and such choice of
the input forcing is not needed. However, other modelling choices have to be made. For instance, several parameterizations are
used to represent ice sheet surface melt (Robinson et al., 2010), like the Positive Degree Day (PDD) method (Reeh, 1991), in
which surface melt depends solely on air temperature, or the Insolation Temperature Melt (ITM) method (Van den Berg et al.,
2008), which takes into account the effect of both temperature and insolation. In both cases, the translation of insolation local
and seasonal variations into ice sheet changes and ice age cycles remains an open modelling question.

The obtention of 100 kyr cycles is not possible with a linear theory like the one of Milankovitch (Hays et al., 1976; Imbrie
and Imbrie, 1980), and a form of non-linearity is needed. Indeed, there is no simple relation between insolation extrema and
ice volume extrema. One of the largest deglaciation (termination V) occured when insolation variations were minimal. On
the contrary, insolation variations where maximal at termination III, whereas the transition was rather small. In addition, the
amplitude of summer insolation variations is maximum every 400 kyr, but this frequency is absent from the paleoclimatic
records. The 100 kyr cycles have been proposed to be linked to either eccentricity driven variations of precession (Raymo,
1997; Lisiecki, 2010), obliquity (Huybers and Wunsch, 2005; Liu et al., 2008), or both (Huybers, 2011; Parrenin and Paillard,
2012), to internal oscillations phase locked to the astronomical forcing (Saltzman et al., 1984; Paillard, 1998; Gildor and
Tziperman, 2000; Tziperman et al., 2006), to internal oscillations independent of the astronomical forcing (Saltzman and
Sutera, 1987; Toggweiler, 2008) or to period doubling bifurcation (Verbitsky et al., 2018). Additionally, the Mid Pleistocene
Transition (MPT) and the switch from a 41 kyr dominated record to a 100 kyr one remains mostly unexplained.

Several conceptual models have been developed to try to solve these questions. Calder (1974) was the first to develop a
simple conceptual model, linking insolation and ice volume variations. Imbrie and Imbrie (1980) also developed a conceptual
model, where the rate of change was different in the case of a warming or cooling climate. Paillard (1998) developed the
idea that the glacial-interglacial cycles can be seen as relaxation oscillations between multiple equilibria, like a glaciation and
a deglaciation state. It was suggested that the criteria to trigger a deglaciation depends on both, insolation and ice volume,
whereas the insolation alone seems able to trigger a glaciation (Parrenin and Paillard, 2003). Here, we adapt and simplify the
model of Parrenin and Paillard (2003) and extend it over the whole Quaternary.

One of the critical questions for conceptual models is to decide which insolation to use as input. In his work, Milankovitch
used "caloric seasons" at 65° N, the half year with the highest insolation. This was also the case in Calder's model, who
used caloric seasons at 50° N as input. Imbrie and Imbrie (1980) used the July insolation. The use of summer insolation
gradually shifted towards the use of the summer solstice insolation, most probably as it is easier to compute thanks to the tables
provided by Berger (1978) and allows to obtain better fits on the more recent part of the records (Paillard, 2015). More recently,
Huybers (2006) suggested that the Integrated Summer Insolation (ISI) over a certain threshold could be better, as it would more

closely follow positive degree days, an important metric for ice sheet mass balance. Others have also used combinations of orbital parameters as a forcing (Imbrie et al., 2011; Parrenin and Paillard, 2012). However, most models only use one type of insolation forcing and do not consider the influence of other insolation forcing on the model results.

In our work, we will consider several summer insolation forcings at 65° N (the summer solstice insolation, the caloric season, and the ISI over two different threshold values) in order to study their influence on the model results. These different summer insolation forcings have similar shape, but the respective contribution of obliquity and precession differ. The different insolations have different performances in matching the data, depending on the time period considered. However, we show that some features are independent of the insolation forcing used as input. In particular, we are able to reproduce a switch from 41 kyr oscillations before the MPT to 100 kyr cycles afterwards in agreement with the records for all insolation forcings, by varying a single parameter : the deglaciation threshold $V_0$, and keeping all the other model parameters constant. This is similar to the results of Paillard (1998) who obtained a frequency shift on the glacial cycles by using a linearly increasing deglaciation threshold. This is also coherent with the more recent results of Tzedakis et al. (2017), which demonstrated that the particular sequence of interglacials that happened over the Quaternary and the frequency shift from 41 to 100 kyr could be explained with a simple rule, taking into account a deglaciation threshold that increases over time, leading to "skipped" insolation peaks and longer cycles.

## 2  Conceptual model and methods

### 2.1  Conceptual model

The model used in our study is an adapted and simplified version of the conceptual model of Parrenin and Paillard (2003). For the glacial-interglacial cycles, it is not a new idea that the climate system can be represented by relaxation oscillations between multiple equilibria, like a glaciation and a deglaciation state (Paillard, 1998; Parrenin and Paillard, 2003, 2012). The aim of conceptual models is not to explicitly model and represent physical processes but rather to help us understand critical aspects of the climate system. Here, we do not intend to explicitly represent the numerous physical processes involved in ice sheet volume evolution, affecting surface mass balance, ice discharge to the ocean and bottom melt of grounded ice. Instead, we represent the climate system by two distinct states of evolution : the "glaciation state" ($g$) and "deglaciation state" ($d$). We make the assumption that the evolution of the ice sheet volume in these two states can be simply described by two terms. The first one, common to the glaciation and deglaciation states, is a linear relation to the summer insolation : when the insolation is above average, the ice sheet melts, whereas when the insolation is low enough, the ice sheet grows. The second term, specific to the system state, represents an evolution trend linked to the system state : a slow glaciation in ($g$) state and a rapid deglaciation in ($d$) state.

In our model, the evolution of the ice volume in these two states is described by :

$$
\begin{cases}
(g) \ \frac{dv}{dt} = -\frac{I}{\tau_i} + \frac{1}{\tau_g} \\
\\
(d) \ \frac{dv}{dt} = -\frac{I}{\tau_i} - \frac{v}{\tau_d}
\end{cases}
\tag{1}
$$

where $v$ represents the normalized ice volume. $\tau_i$, $\tau_d$ and $\tau_g$ are time constants. $I$ is the normalized summer insolation forcing at 65° N. The implicit assumption is made that the global ice volume changes are mainly driven by the Northern Hemisphere ice sheet waning and waxing, as we focus on the effect of insolation changes at high northern latitudes. This is not a limiting assumption as data suggest much lower sea level variation due to the Antarctic ice sheet than the Northern Hemisphere ones over glacial-interglacial timescales. For example, the contribution of the Antarctic ice sheet to the ∼120 m Last Glacial Maximum sea level drop is estimated to be between 10 and 35 m sea level equivalent (Lambeck et al., 2014). Other effects like thermal expansion, small glaciers and ice caps are estimated to be around 3 to 4 m sea level equivalent. Furthermore, it has been suggested (Jakob et al., 2020) that the growth of larger Northern Hemisphere ice sheets since the start of the Quaternary and the associated sea level drop has a stabilizing effect on the East Antarctic ice sheet, as it limits its exposure to warm ocean waters.

A critical point is to define the criteria for the switch between the glaciation and deglaciation states. To enter the deglaciation state, both ice volume and insolation seem to play a role (Raymo, 1997; Parrenin and Paillard, 2003, 2012), as terminations occur after considerable build-up of ice sheet over the last million year. To represent the role of both ice volume and insolation in the triggering of deglaciations, the condition to switch from $(g)$ to $(d)$ state uses a linear combination of ice volume and insolation. The deglaciation is triggered when the combination crosses a defined threshold $V_0$ : the deglaciation threshold. As in the work of Parrenin and Paillard (2003), this allows transitions to occur with moderate insolation when the ice volume is large enough and reciprocally. On the contrary, glacial inceptions seem to depend on orbital forcing alone (Khodri et al., 2001; Ganopolski and Calov, 2011). Therefore, the condition to switch from the deglaciation state to the glaciation state is based on insolation only : it is possible to enter glaciation when the insolation becomes low enough.

$$
\begin{cases}
(d) \ to \ (g) \ : I < I_0 \\
\\
(g) \ to \ (d) \ : I + v > V_0
\end{cases}
\tag{2}
$$

The idea that the deglaciation threshold is linked to both, insolation and ice volume, is not new (Parrenin and Paillard, 2003, 2012), and is similar to the one developed by Tzedakis et al. (2017), where the threshold for a complete deglaciation decreases with time as the system accumulates instability with ice sheets becoming more sensitive to insolation increase. As the ice sheet grows and extends to lower latitudes, the insolation needed to reach a negative mass balance decreases. This idea has been confirmed with modelling studies (Abe-Ouchi et al., 2013). Several reasons can explain this increase of instability over time : delayed bedrock rebound and exposure to higher temperature as the ice sheet sinks (Oerlemans, 1980; Pollard, 1982; Abe-Ouchi et al., 2013), increase in basal sliding as the ice sheet grows and the base becomes warmer, as more isolated from

the cold surface temperature (MacAyeal, 1992; Marshall and Clark, 2002), ice-sheet margins calving (Pollard, 1982), decrease of the ice sheet albedo due to either snow aging (Gallée et al., 1992) or increase of dust deposition as the ice sheet expands (Peltier and Marshall, 1995; Ganopolski and Calov, 2011), increase of atmospheric $CO_2$ due to release of deep ocean carbon when the Antarctic ice sheet extends over continental shelves (Paillard and Parrenin, 2004; Bouttes et al., 2012).

## 2.2 Summer insolation

The conceptual model defined previously uses summer insolation as input. It is therefore important to consider which insolation should be used. Insolation is usually taken at 65° N, a typical latitude for Northern Hemisphere ice sheets. Several articles have used the summer solstice daily insolation at 65° N as input insolation forcing (Paillard, 1998; Parrenin and Paillard, 2003). Others (Tzedakis et al., 2017) have used the caloric season, as defined by Milankovitch (1941), defined as the half year with the highest insolation. It is also possible to use as input a linear combination of orbital parameters (Imbrie et al., 2011; Parrenin and Paillard, 2012). Huybers (2006) also defined the Integrated Summer Insolation (ISI) above a threshold.

Here, for the first time, we want to examine the effect of this choice on the dynamics of a conceptual model. We therefore use four different summer insolation forcings, and compare the model results obtained with each of them. We use the summer solstice insolation, the caloric season, and the ISI above two different thresholds (300 W $m^{-2}$ and 400 W $m^{-2}$). Experiments were also conducted for the summer solstice insolation at 50° N instead of 65° N but are not presented here as they do not change the conclusions obtained with the forcings at 65° N. We refer the reader to Sect. 1 in the Supplement for these experiments.

These four insolation forcings differ, and have different contributions from precession and obliquity, as can be seen in Fig. 1. For example, the summer solstice insolation has a low contribution from obliquity (41 kyr) compared to the caloric season, or the ISI above 300 W $m^{-2}$. The summer solstice insolation and the caloric season were computed using the Analyseries software (Paillard et al., 1996). The ISI for a threshold $\tau$ (noted as ISI above $\tau$ or ISI($\tau$) in this article) was defined by Huybers (2006) as the sum of the insolation on days exceeding this threshold.

$$ISI(\tau) = \sum_i \beta_i (W_i \cdot 86400)$$

Where $W_i$ is the mean insolation of day $i$ in W $m^{-2}$ and $\beta_i$ equals one if $W_i > \tau$ and zero otherwise.

To compute $ISI(\tau)$, one first needs to compute the daily insolation on day $i$, and then sum over the year for the days exceeding the threshold. Here, we developed a Python code based on the Matlab code provided by Huybers (2006). Unlike the insolation files provided by Huybers (2006), which used the solution from Berger and Loutre (1991), we use the orbital parameters value of the Laskar et al. (2004) solution for the calculation. This results in slightly different results for deeper time periods as the calculation of orbital parameters also differ with these two estimations.

## 2.3 Optimal model parameters

The conceptual model relies on a small number of model parameters : $\tau_i$, $\tau_g$, $\tau_d$ and the two thresholds for the switch from a glaciation state to a deglaciation state and inversely : $V_0$ and $I_0$. For all the simulations performed, we kept $\tau_i = 9$ kyr, $\tau_g =$

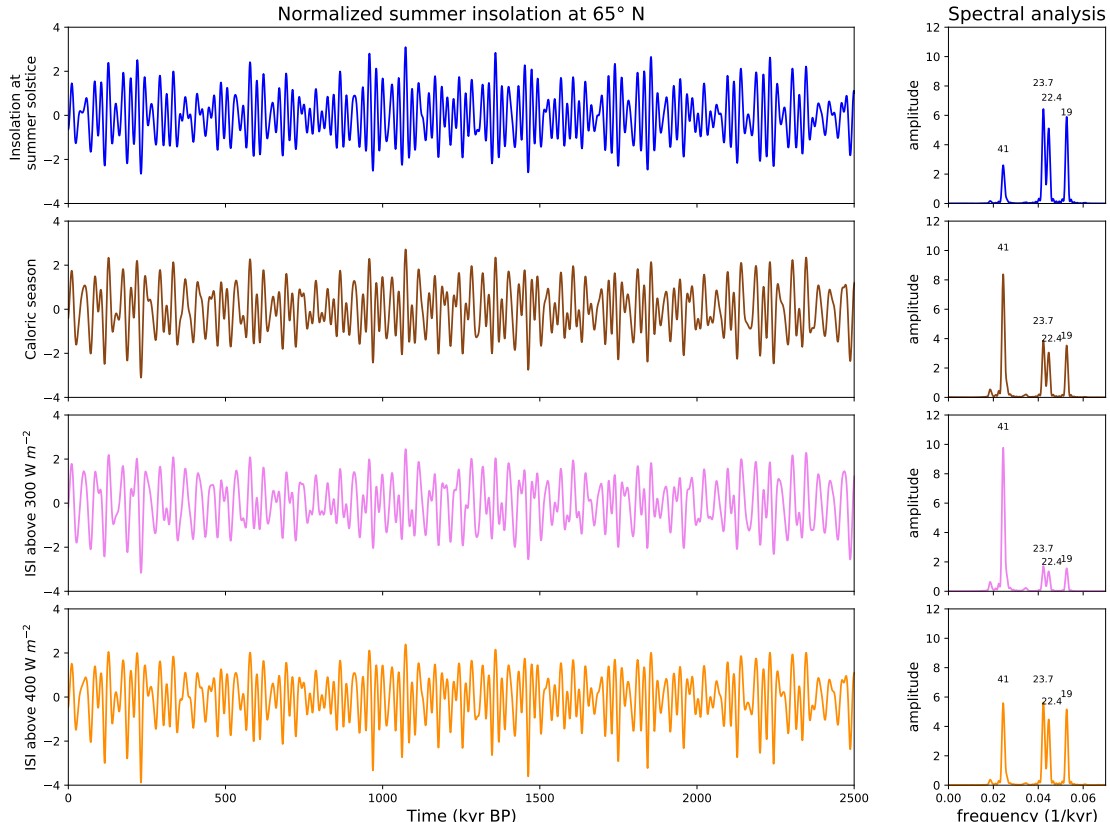

**Figure 1.** (a) The four different summer insolation forcings at 65° N (summer solstice, caloric season, ISI above 300 W $m^{-2}$ and above 400 W $m^{-2}$) displayed over the Quaternary (respectively in blue, brown, pink and orange). The insolation forcings are centered and normalized by their standard deviation. (b) Corresponding spectral analysis over the Quaternary, normalized by standard deviation.

30 kyr, $\tau_d$ = 12 kyr and $I_0$ = 0, as these parameters gave correct behaviour in previous studies (Parrenin and Paillard, 2003). No attempt was made to tune these parameters to have a behaviour closer to the data. On the contrary, more focus was set on the influence on the model results of the deglaciation threshold parameter, $V_0$. To compare our model results to data, we used the benthic $\delta^{18}O$ stack "LR04" (Lisiecki and Raymo, 2005) as a proxy for ice volume, considering that most of the $\delta^{18}O$ changes of benthic foraminifera represent changes in continental ice (Shackleton, 1967; Shackleton and Opdyke, 1973). Lower $\delta^{18}O$ values correspond to lower ice volume. The model results as well as the LR04 curve were normalized to facilitate their comparison. In the following "data" refers to the $\delta^{18}O$ stack curve LR04 normalized.

To estimate which model parameters lead to model results closer to the data, the definition of an objective criteria is needed. The choice of such a criteria is not straightforward, and the use of different criteria, could have led to slightly different results. Our model is simple and does not aim at reproducing precisely the ice volume evolution, but rather at reproducing the main qualitative features, such as the shape and frequency of the oscillations. Therefore, we used a criteria based on the state of the system : glaciation or deglaciation. Similar results can be obtained using a simple correlation coefficient (see Sect. 2 in the Supplement). The definition of the deglaciation state in the data is explained in Sect. 2.4. A critical point is that our model should be able to deglaciate at the right time, when a deglaciation is seen in the data. Conversely, the model should not produce deglaciation at periods where the data do not show deglaciation.

We defined a criteria for each of these two conditions, and assembled them in a global criteria. To determine if a deglaciation is well reproduced by our model, we look at the state of the model (glaciation or deglaciation) at the time halfway between the start of the deglaciation and the end of the deglaciation. If the model state at that time is deglaciation, the deglaciation is considered as correctly reproduced. Otherwise, it is considered as a "missed" deglaciation. We simply defined the criteria $c_1$ as the fraction of deglaciations correctly reproduced. This criteria equals to one when all the deglaciations take place at the right time, meaning that the model produces a deglaciation state every time a deglaciation is seen in the data. To ensure that the model does not deglaciate too often, we looked at insolation maxima, that are not associated with deglaciation in the data and ensured the corresponding model state was glaciation. We decided to look specifically at insolation maxima, as it is where the model is the most likely to deglaciate when it should not. A deglaciation seen in the model at a place where no deglaciation is seen in the data is considered as a "wrong transition". We defined the criteria $c_2$ as $c2 = 1 - w$, with $w$ the fraction of wrong transitions. The $c_2$ criteria equals to one when the model does not deglaciate when it should not. To take into account these two conditions, the overall accuracy criteria $c$ is defined as $c = c_1 \cdot c_2$. It is equal to one when all the deglaciations are correctly placed and that no additional deglaciation compared to the data take place.

In order to study the evolution of the optimal deglaciation threshold $V_0$ over the Quaternary, it was divided into five 500 kyr periods. The $V_0$ values that maximize the accuracy criteria for each time period and insolation forcing are called "optimal $V_0$". To determine the optimal $V_0$ threshold corresponding to each period and insolation forcing, several simulations were carried out and the parameter values maximizing the accuracy criteria c were selected. More precisely, for each insolation and period, 3500 simulations corresponding to different $V_0$ thresholds (from $V_0 = 1.0$ to $V_0 = 8.0$ with a step of 0.1) and different initial conditions (initial volume $V_{init}$ ranging from 0.0 to 5.0 with a step of 0.2, and the two possible initial states - glaciation or deglaciation) were performed. The numerical integration of the model equations was done with a fourth order Runge Kutta scheme.

For each insolation forcing, the best fit over the Quaternary is defined as the simulation over the whole Quaternary (0 to 2500 ka BP) with a $V_0$ changing with time, that is equal to the corresponding optimal $V_0$ at each time period. Additionally, simulations were performed to determine the optimal $V_0$ threshold obtained when the optimization procedure is carried out over the whole Quaternary. It is called $V_0^Q$ in the following.

## 2.4 Definition of the deglaciation state in the data

To calculate our accuracy criteria $c$ and therefore determine the optimal $V_0$ threshold over a given period, a definition of the deglaciation in the data is needed. We based our definition on the interglacial classification developed by Tzedakis et al. (2017). Tzedakis et al. (2017) differentiates integlacials, continued interglacials and interstadials based on a detrended version of the LR04 stack curve. A period is considered as an interglacial if its isotopic $\delta^{18}O$ is below a threshold (3.68 ‰). Two interglacials are considered as separated if separed by a local maximum above a threshold (3.92 ‰). This definition differs from the usual characterization of terminations, sometimes leading to several interglacials in the same isotopic stage. The definition of Tzedakis et al. (2017) is for interglacials, and as our focus is not on interglacial periods but rather on deglaciations, we adapted it. We defined deglaciations as periods of decreasing $\delta^{18}O$ (and thus, ice volume, in our assumptions) preceeding the interglacials. The start of the deglaciation is taken as the last local maxima above the threshold of 3.92 ‰, and the end of the deglaciation is taken as the first local minima below the interglacial threshold of 3.68 ‰. The deglaciation periods in the data corresponding to the time between the deglaciation starts and deglaciation ends are displayed with a blue shading in Fig. 4 and Fig. 5.

## 3 Results and discussion

### 3.1 Optimal deglaciation threshold $V_0$ and corresponding accuracy

For each insolation, the deglaciation threshold $V_0$ maximizing the accuracy for each of the five 500 kyr periods, as well as the fixed $V_0^Q$ value maximizing the accuracy over the whole Quaternary were computed. The results are displayed in Fig. 2. In some cases, several values of the deglaciation threshold $V_0$ lead to the same accuracy criteria, whereas in other there is only one $V_0$ value maximizing the accuracy. When several $V_0$ values are equivalent, the mean value was plotted and the other possible values are represented with errorbars.

Over the same time period, different insolation values lead to slightly different optimal $V_0$ threshold. But the most striking feature, is the increase of the optimal $V_0$ threshold over time, and more specifically for the last million year, after the Mid Pleistocene Transition. This feature is valid whatever the insolation type. The optimal $V_0$ over the whole Quaternary $V_0^Q$ is between 3.4 and 4 for each insolation type. It is a value in the middle of the highest values that best fit the latter part of the record and the lowest values that best fit the earliest part of the record.

For each insolation, the accuracy corresponding to the optimal $V_0$ threshold for each time period as well as to the fixed $V_0^Q$ value maximizing the accuracy over the whole Quaternary is displayed in Fig. 3. It is first noticeable, that the accuracy is higher on the last million year period, whatever the input summer insolation used. On the last million year, the summer solstice insolation as input produces the best results. However, this is not the case anymore for more ancient time periods. On the contrary, the solstice insolation gives the worst results at the start of the Quaternary. The accuracy obtained on the whole Quaternary period (fixed $V_0^Q$ value) is globally lower than the accuracy on each time period. This is due to the fact that the $V_0^Q$

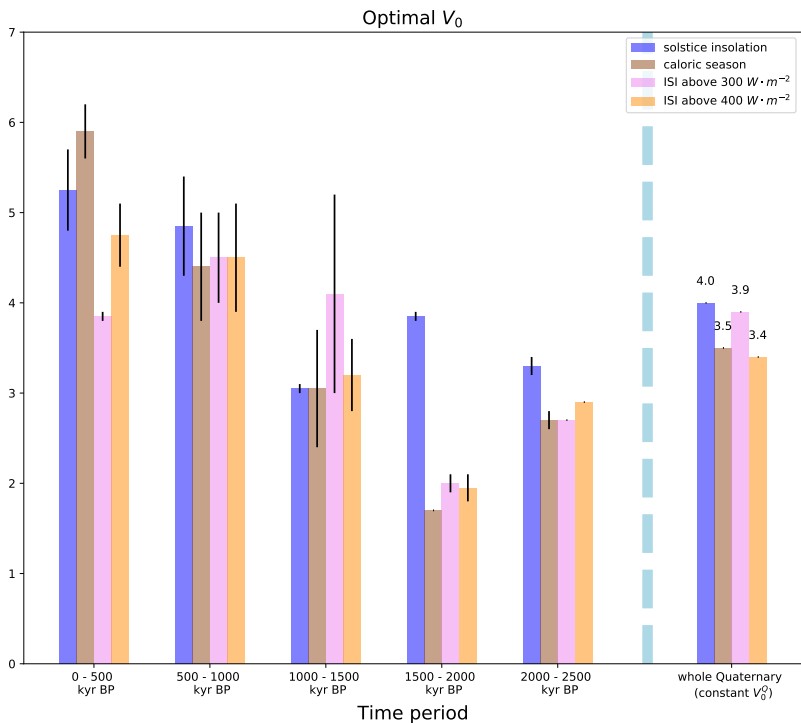

**Figure 2.** Optimal deglaciation threshold $V_0$ over the five 500 kyr periods for the four different summer insolation forcings at 65° N and optimal constant $V_0$ threshold obtained when the optimization procedure is done over the whole Quaternary ($V_0^Q$). When several values of the deglaciation threshold $V_0$ maximize the accuracy criteria, the mean value is plotted and the other possible values are represented with errorbars.

values obtained are lower than the optimal $V_0$ values on the later part of the Quaternary and higher than the optimal $V_0$ values on the earliest part of the Quaternary, leading to a poorer representation of both of these periods.

On the earlier part of the Quaternary (periods earlier than 1.5 Ma BP), the results are less robust. This is due to increased incertainties in the LR04 record, and the associated definition of interglacial periods, which affects our accuracy criteria. In their classification, Tzedakis et al. (2017) stated that the definition of the valley depth needed to separate several interglacials was quite straighforward on the earlier part of the record, whereas for earlier time periods (ages older than 1.5 Ma BP), their method led to several "borderline cases", and that slighlty different choices for the interglacial and interstadial thresholds would have led to a different population of interglacial. For us, this would lead to a different definition of deglaciation, and thus a different accuracy criteria. In addition, the resolution of the LR04 curve decreases with increasing age (Lisiecki and Raymo, 2005), and for periods with lower resolution or more uncertain age matching, the amplitude of the peaks might be reduced (Tzedakis et al., 2017).

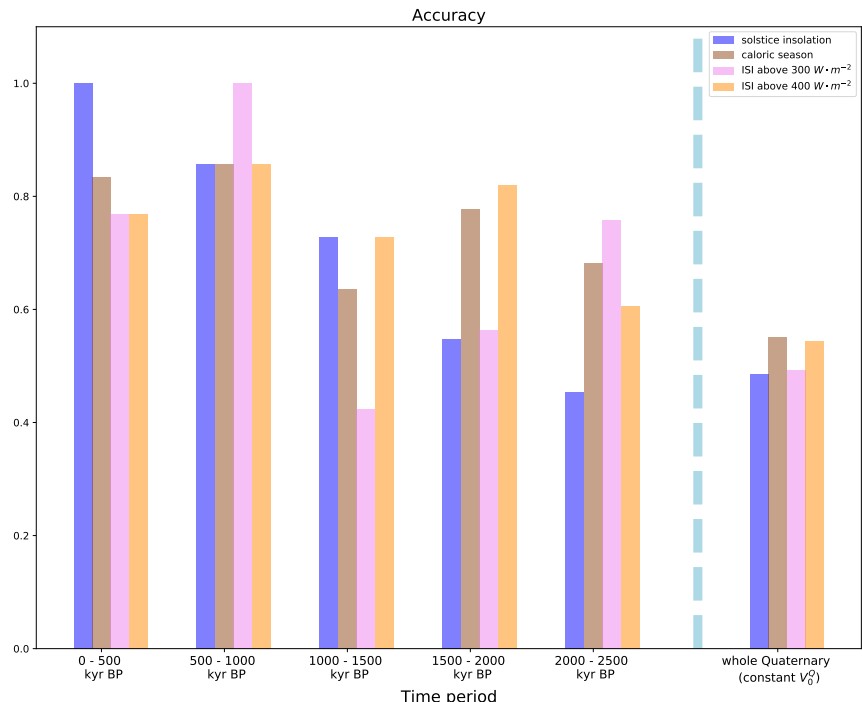

**Figure 3.** Accuracy over the five 500 kyr periods for the four different summer insolation forcings at 65° N, and accuracy criteria over the Quaternary when the optimization procedure is done over the whole Quaternary (constant $V_0^Q$).

Moreover, the $\delta^{18}O$ LR04 curve includes at the same time an ice volume and deep water temperature component. Ice volume and sea level reconstructions do exist (Bintanja et al., 2005; Spratt and Lisiecki, 2016), but are however limited to the more recent part of the Quaternary and do not allow the investigation of the pre MPT period. The use of $\delta^{18}O$ as an ice volume

proxy has already been largely debated (Shackleton, 1967; Chappell and Shackleton, 1986; Shackleton and Opdyke, 1973; Clark et al., 2006) and recent studies (Elderfield et al., 2012) have shown that the temperature component may be as large as 50%. Furthemore, the stack was tuned to insolation (Lisiecki and Raymo, 2005). We refer the reader to (Raymo et al., 2018) for a review of possible biases in the interpretation of the LR 04 benthic $\delta^{18}O$ stack as an ice volume and sea level reconstruction. All these reasons encourage us to remain at a qualitative level to fit the data.

The increase in the optimal deglaciation threshold $V_0$ over the MPT is however a robust feature, that does not depend on the input summer insolation forcing used. This is coherent with the work of Paillard (1998) and Tzedakis et al. (2017) who obtained a frequency shift from 41 kyr cycles to 100 kyr cycles with an increasing deglaciation threshold. Classical hypothesis and possible physical meanings of the rise of the $V_0$ threshold often imply a gradual cooling, due to lower $pCO_2$ (Raymo, 1997; Paillard, 1998; Berger, 1978), but this idea is not supported by more recent data, which do not show a gradual decrease

in $pCO_2$ trend, but rather only a decrease in minimal $pCO_2$ values (Hönisch et al., 2009; Yan et al., 2019). Other hypothesis

imply a gradual erosion of a thick regolith layer, that exposed the ice sheet to a higher friction bedrock allowing thicker ice sheet to develop (Clark and Pollard, 1998; Clark et al., 2006), or sea ice switching mechanism (Gildor and Tziperman, 2000).

The overall lowest accuracy on the more ancient part of the record suggests that our non linear threshold model is less adapted for this time period. Indeed, the ice volume might respond more linearly to the insolation forcing before the MPT, as some studies suggest (Tziperman and Gildor, 2003; Raymo and Nisancioglu, 2003). In contrast, others (Ashkenazy and Tziperman, 2004) have shown that the 41 kyr pre MPT oscillations are in fact significantly asymmetric and therefore suggested that oscillations both before and after the MPT could be explained as self sustained variability of the climate system, phase locked to the astronomical forcing. In our model, the deglaciation threshold $V_0$ changes over time leading to an amplitude change of the cycles. However, it cannot be excluded that the mechanisms behind the pre and post MPT glacial cycles are structurally different, and that they cannot be explained with the same physics and the same equations in our model. Moreover, this kind of conceptual model has mainly been constructed in order to explain the non linear feature of the 100 kyr cycles, and it is therefore not surprising that its agreement with the data on the pre MPT period is less satisfying.

## 3.2 Best fit over the Quaternary

Our conceptual model is able to reproduce qualitatively well the data (LR 04 normalized curve) over the whole Quaternary. The model's best fit over the Quaternary for each insolation forcing, as defined in Section 2.3, is displayed in Fig. 4. It is able to reproduce the frequency shift from a dominant 41 kyr period before 1 Ma BP to longer cycles afterwards, as observed in the data, and thus by varying only one parameter during the whole simulation length : the deglaciation threshold $V_0$. Like the previous model of Parrenin and Paillard (2003) from which our model is adapted, ours looses quicky its sensitivity to the initial conditions (after no more than 200 kyr). For every input forcing, longer cycles are obtained on the last part of the Quaternary (last Myr). Figure 4 displays the results over the whole Quaternary with the $V_0$ threshold being set to its optimal value on each 500 kyr period, while Figure 5 displays the results over the last million year with the $V_0$ threshold being set to its optimal value on the [0 - 1000] ka BP period.

For the last million year, it is possible to reproduce with the right timing all terminations, apart from the last deglaciation, for all insolation forcings, with a single value of the $V_0$ threshold over this period. Some differences are however noticeable between the different forcings. Especially for the ISI above 300 W $m^{-2}$ forcing, the agreement is not as good as for the other forcings : Termination V (around 420 ka BP) is triggered later compared to the data, while Termination III (around 240 ka BP) is triggered too early. For the ISI above 300 W $m^{-2}$ forcing, the range of $V_0$ values allowing to reproduce correctly most of the terminations on the last million year is reduced (only values of $V_0 = 3.9$ - 4.0), whereas the results are more robust for the three other insolations forcings, with a broader range of working $V_0$ values. The ISI above 300 W $m^{-2}$ forcing has a low precession component, which explains why it is less successful in reproducing the data over the last million year. Experiments with our model setup have shown that a summer forcing with no precession component could not successfully reproduce the data over the post MPT period as accurately as the four forcings presented here, that contain both precession and obliquity (see Sect. 3 in the Supplement).

Despite the accurate timing of terminations, the spectral analysis of the model results over the last million year differs from the spectral analysis of the data. For all forcings except the summer solstice insolation, obliquity continues to dominate after the MPT. The spectral analysis shows secondary and third peaks of lower frequency, but does not show a sharp 100 kyr cyclicity as in the LR04 record. Compared to the data, all the model outputs over the post MPT period have a more pronounced obliquity and precession component and a less pronounced 100 kyr component. This feature is most probably due to the model formulation, and more specifically the direct dependence of ice volume evolution to insolation via the $dV/dt = -I/\tau_i$ term. This is one of the limits of our conceptual model. While the criteria on the switch to deglaciations allows us to reproduce the deglaciations at the right timing, the direct dependance of ice volume change to the insolation forcing is definitely too simplistic and probably produces an overestimated dependency of the ice evolution to the astronomical forcing on the latter part of the record.

On the first part of the Quaternary (2.6 Ma BP to 1 Ma BP), the spectral analysis of the data is dominated by a 41 kyr (obliquity) peak. It is also the case for the model results, for each type of insolation. However, the model outputs also show a precession component (19 to 23 kyr), especially for the summer solstice and the ISI above 400 W $m^{-2}$ forcings, which does not exist on the data. Although our model reproduces qualitatively well the feature of the record, there are some noticeable model data mismatch that occur for all insolation type, around 1100 ka BP and 2030 ka BP. Additionally, the amplitude of the oscillations produced is slightly too high on the more ancient part of the Quaternary, especially when the ISI above 300 W $m^{-2}$ is used. On the oldest part of the Quaternary, the caloric season forcing and the ISI over 300 W $m^{-2}$ seem to perform better in reproducing the frequency of the oscillations than the summer solstice insolation and ISI over 400 W $m^{-2}$. They correspond to the forcing with the strongest obliquity (41 kyr) component, which might explain why they more successfully represent this part of the record, that is dominated by obliquity.

Over the last million year, the highest accuracy is obtained with the summer solstice insolation as input forcing ($c = 0.92$ for the summer solstice insolation, $c = 0.82$ for the caloric season and Integrated Summer Insolation above 400 W $m^{-2}$ and $c = 0.87$ for the Integrated Summer Insolation above 300 W $m^{-2}$). This is mainly due to the fact that for the three other forcings (caloric season and ISI above 300 W $m^{-2}$ and 400 W $m^{-2}$), the last deglaciation occurs one insolation peak too early, around 50 ka BP. However, if one computes the accuracy over the last million year excluding the last 100 kyr (period from 100 ka to 1000 ka BP), the accuracy is similar for all four forcings (0.92 for summer solstice, caloric season and ISI above 400 W $m^{-2}$ and 0.89 for ISI above 300 W $m^{-2}$). Figure 5 displays two different alternatives. For the first one (full line), the optimal $V_0$ is calibrated over the [100 - 1000] ka BP period and maintained for the whole simulation ($V_0 = 5.0$ for the summer solstice insolation, 4.5 for the caloric season, 4.0 for the ISI above 300 W $m^{-2}$ and 4.6 for the ISI above 400 W $m^{-2}$). Except with the summer solstice insolation as input, the three other insolation forcings fail to reproduce accurately the last deglaciation, as the last deglaciation occurs one insolation peak too early. The second alternative (dashed line) is to raise the deglaciation threshold $V_0$ over the last cycle (raised to $V_0 = 5.5$ from 100 ka BP onward). In this case, the model does reproduce accurately the last deglaciation for all insolation forcings. This suggest, that we might need to raise further the $V_0$ threshold in order to model a theoretical "natural" future (without anthropogenic influence) with longer cycles. A cycle is not enough to conclude on a trend,

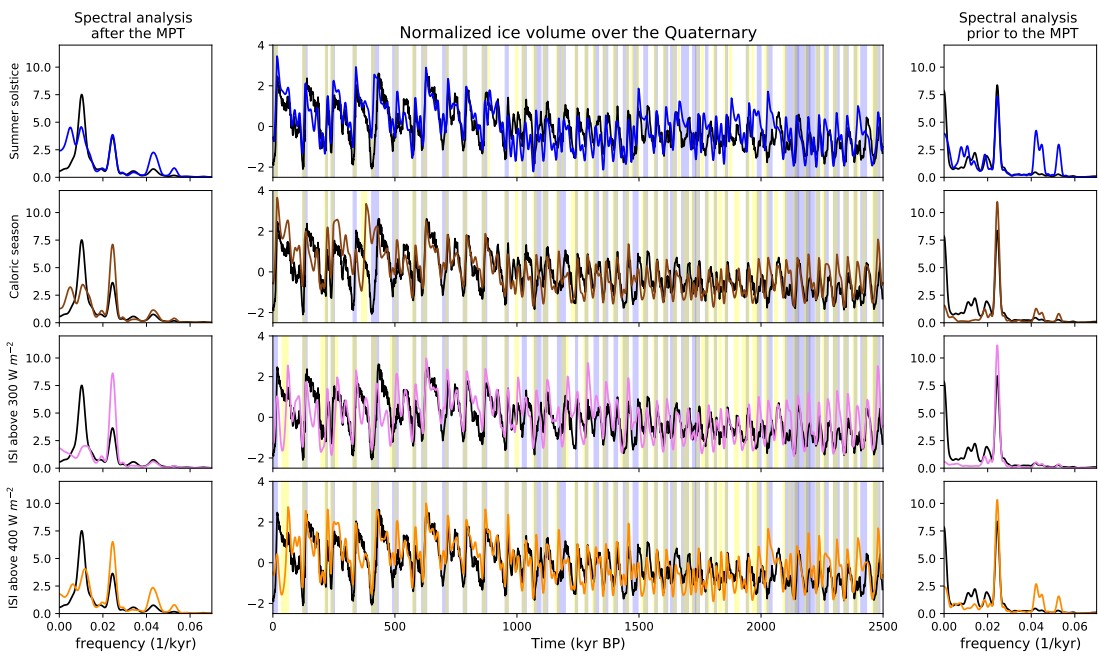

**Figure 4.** Best model fit over the whole Quaternary and corresponding spectral analysis. The middle panel represents the best fit of the model for the different summer insolation used as input, compared to the data. The results for the summer solstice insolation, caloric season, ISI above 300 W $m^{-2}$ and 400 W $m^{-2}$ are respectively displayed in blue, brown, pink and orange. The data (normalized LR04 curve) are in black. The blue shading represents deglaciation periods in the data and the yellow shading deglaciation periods in the model. This results in a green shading when deglaciations are seen in the model and data at the same time. The left panel represents the spectral analysis of the best fit solution over the last million year. The right panel represents the spectral analysis over the more ancient part of the Quaternary (before 1 Ma BP).

but this should be envisaged for future natural scenarios. This is coherent with the idea of Paillard (1998) who used a linearly increasing deglaciation threshold over the Quaternary.

## 3.3 Reflexions about the future

To model future natural evolutions of the climate system, possible evolutions of the $V_0$ threshold should be considered. However, we do not exclude the fact that variations of other parameters, that were kept constant in this study, could vary in the future. For instance, different $I_0$ thresholds have to be considered. The fate of the next kyrs in a natural scenario cannot be ascertained with our conceptual model. Indeed, it depends on the onset (or not) of a glaciation, which is determined by the $I_0$ parameter. In our model, a broad range of $I_0$ parameter values lead to satisfying results over the last million year. These

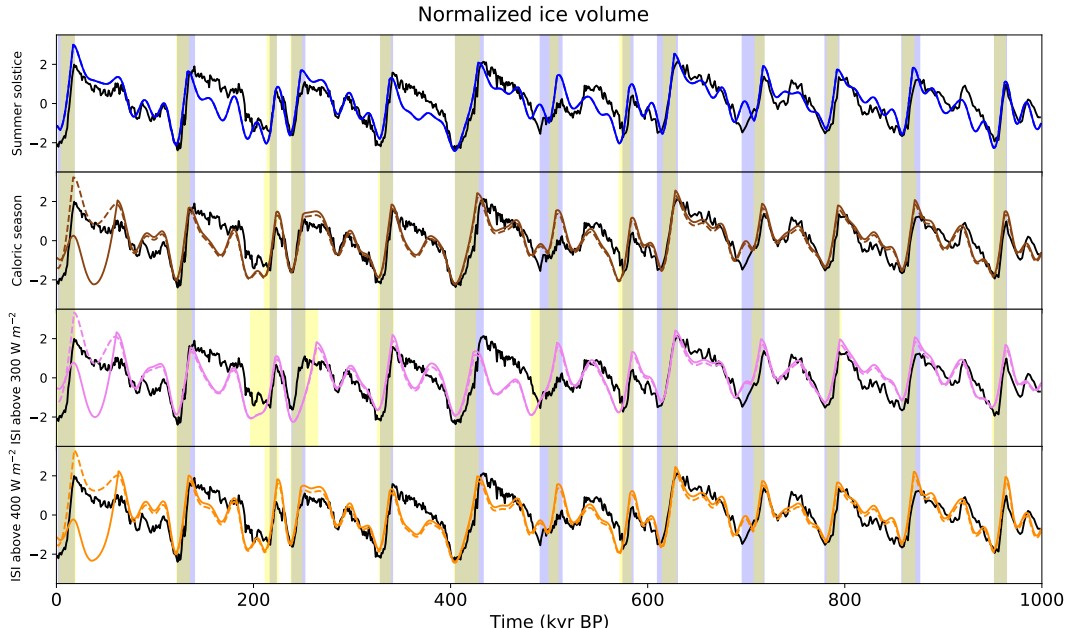

**Figure 5.** Normalized model results over the last million year, with the different summer insolation forcings : insolation at the summer solstice (blue), caloric season (brown), ISI above 300 W $m^{-2}$ (pink), and over 400 W $m^{-2}$ (orange). The full line is the best fit computed over the 100 - 1000 ka BP period. The dashed line represents the same solution, but with an increased $V_0$ threshold for the last deglaciation. The data (LR04 stack curve normalized) are in black. The blue shading represents deglaciation periods in the data. The yellow shading represents deglaciation period in the model outputs (case of the increased $V_0$ threshold). This results in a green shading when deglaciations are seen in the model and data at the same time.

different values projected into the future either lead in one case to a glaciation start at present, and in the other case in a continued deglaciation, switching to a glaciation state only around 50 kyr after present at the next insolation minima. This is due to the current particular astronomical configuration, with a very low eccentricity, which leads to high summer insolation minima, where the threshold for glaciation might not be reached. This is coherent with previous studies (Berger and Loutre, 2002; Paillard, 2001) which suggested that without anthropogenic forcing, the present interglacial might have lasted 50 kyrs.

However, this exercise is purely academic, as we are not taking into account the role of anthropogenic $CO_2$, which would affect the glaciation and deglaciation thresholds (Archer and Ganopolski, 2005; Talento and Ganopolski, 2021). Furthermore, our conceptual model cannot be extended outside the Quaternary, as the ice volume variations considered are exclusively the Northern Hemisphere one, and our model is, by construction unable to represent projected future Antarctic ice sheet mass loss.

## 4 Conclusions

We have used a conceptual model with very few tunable parameters that represents the climatic system by multiple equilibria and relaxation oscillation. Only one parameter was varied, the deglaciation threshold parameter $V_0$. We used different summer insolation as input for our conceptual model : the summer solstice insolation, the caloric season, and the Integrated Summer Insolation over two different thresholds. With all these forcings, that have different contributions from obliquity and precession, we are able to reproduce the feature of the ice volume over the whole Quaternary. More specifically, we are able to represent the Mid Pleistocene Transition and the switch from a 41 kyr dominated record to larger cycles, by raising the deglaciation threshold and keeping the other model parameters constant. This rise in the deglaciation threshold is valid regardless of the type of summer insolation forcing used as input. However, the data agreement is less satisfying before the MPT. This suggests the possibility that climate mechanisms might be structurally different before and after the MPT, with a more linear behaviour in pre-MPT conditions. This highlights that models are designed to answer rather specific questions, and a model built specifically to explain 100 kyr cycles might be less efficient in a more linear setting. More generally, this kind of glacial-interglacial conceptual model is designed to explain the main features of the Quaternary time period characterized by the waning and waxing of Northern Hemisphere ice sheets under the influence of changing astronomical parameters. In our case, this raises the question of what physical phenomena are responsible for making deglaciations "harder" to start on the latest part of the Quaternary compared to the earliest part. This kind of model is however unlikely to be directly applicable in a more general context, like the Pliocene and earlier periods, or in the context of future climates under the long-term persistence of anthropogenic $CO_2$ (Archer and Ganopolski, 2005; Talento and Ganopolski, 2021). In order to tackle such questions, it would be critical to gain a deeper understanding of the natural evolution of the other forcings involved in the climate system and most notably of the dynamics of the carbon cycle. Conceptual models are likely to pave the way in this direction (Paillard, 2017). Indeed, just as in the case of the Quaternary, a full mechanistic simulation of the many processes at work is currently out of reach and the modelling work can only be very exploratory. Here, we have shown that some robust features are required to explain Quaternary ice age cycles. Similar conceptual modelling on a wider temporal scope over the Cenozoïc could help better understand the connections between the astronomical forcing, the carbon cycle, ice sheets and climate. This would help us imagine what the Anthropocene might be like.

*Code and data availability.* The model code as well as the insolation input files, spectral analysis and code needed to reproduce the figures are available for download online : https://doi.org/10.5281/zenodo.6045532

*Author contributions.* GL and DP designed the study. GL performed the simulations, and wrote the manuscript under the supervision of DP.

*Competing interests.* The authors declare that they have no conflict of interest.

*Acknowledgements.* We acknowledge the use of the LSCE storage and computing facilities and thank ANDRA for their financial support.
We also thank the reviewers for their helpful comments and suggestions.

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
