# Peer review of "Influence of the choice of insolation forcing on the results of a conceptual glacial cycle model"

_Climate of the Past, 2021_

## Referee Comment (RC1)

**Review of the manuscript cp-2021-119 "Influence of the choice of insolation forcing on the results of a conceptual glacial cycle model" by Gaëlle Leloup and Didier Paillard**

The motivation for the research is well articulated. Indeed, the insolation amplitudes under different configurations of the orbital forcing may vary significantly enough to potentially influence our understanding of the climate history. To demonstrate the sensitivity of modeling results to the orbital forcing configuration, the authors offer a number of numerical experiments with a specific dynamical model.

Unfortunately, the experimental set-up is not comprehensive enough and therefore the results are not conclusive.

The authors are a bit overoptimistic when they introduce their model as the one having only 5 governing parameters. In fact, in the presented set-up, the model has 13 governing parameters, specifically:

$I_0, V_0, \tau_i, \tau_g, \tau_d, I_{41}, I_{23}, I_{22}, I_{19}, T_{41}, T_{23}, T_{22}, T_{19}$

Here $T_{41}, T_{23}, T_{22}, T_{19}$ are orbital periods (an index shows a numerical value in kyr) and $I_{41}, I_{23}, I_{22}, I_{19}$

are corresponding insolation amplitudes. Only timescales and orbital periods have dimension of time, other parameters are adimensional.

If we select, let say, the obliquity period as a parameter with independent dimension, then, according to $\pi$-theorem, the amplitude of the 100-kyr system response $V_{100}$ will be a function of 12 adimensional similarity parameters:

$$V_{100} = \Phi(I_0, V_0, \frac{T_{41}}{\tau_i}, \frac{T_{41}}{\tau_g}, \frac{T_{41}}{\tau_d}, I_{41}, I_{23}, I_{22}, I_{19}, \frac{T_{41}}{T_{23}}, \frac{T_{41}}{T_{22}}, \frac{T_{41}}{T_{19}}) \tag{R1}$$

It would be physically reasonable to assume $\frac{T_{41}}{T_{23}}, \frac{T_{41}}{T_{22}}, \frac{T_{41}}{T_{19}}$, and $I_0$ to be constant and, using generalized $\pi$-theorem, to migrate to 8 similarity parameters:

$$V_{100} = \Phi(V_0, \frac{T_{41}}{\tau_i}, \frac{T_{41}}{\tau_g}, \frac{T_{41}}{\tau_d}, I_{41}, I_{23}, I_{22}, I_{19}) \tag{R2}$$

The authors demonstrated that period-doubling bifurcation caused by rising deglaciation threshold $V_0$ is not very sensitive to the choice of $I_{41}, I_{23}, I_{22}, I_{19}$. Unfortunately, they stopped here, and, by avoiding variations of $\frac{T_{41}}{\tau_i}, \frac{T_{41}}{\tau_g}, \frac{T_{41}}{\tau_d}$ similarity parameters, they left readers with the impression that changing deglaciation threshold $V_0$ is the only option to reproduce the middle-Pleistocene transition. As Verbitsky et al (2018) and Verbitsky and Crucifix (2020, 2021) have demonstrated, the space of possibilities to produce a period-doubling bifurcation similar to the middle-Pleistocene transition is much wider and (if we continue to speak in terms of the model being reviewed) $\frac{T_{41}}{\tau_i}, \frac{T_{41}}{\tau_g}, \frac{T_{41}}{\tau_d}$ similarity parameters definitely have their roles in the drama. The authors' reasoning to keep $\tau_i, \tau_g, \tau_d$ constant because "these parameters gave correct behaviour in previous studies" is surprising – actually, their past experience with $\tau_i, \tau_g, \tau_d$ speaks about importance of these parameters for the system dynamics and therefore strongly advocates for them to be in the center of the study. Formally, making $\tau_i, \tau_g, \tau_d$ constant is equivalent to claiming complete similarity in parameters $\frac{T_{41}}{\tau_i}, \frac{T_{41}}{\tau_g}, \frac{T_{41}}{\tau_d}$ and excluding them from the solution (R2). Since $\frac{T_{41}}{\tau_i}, \frac{T_{41}}{\tau_g}, \frac{T_{41}}{\tau_d}$ reference values are of the same order of magnitude as $V_0$, there is absolutely no physical justification for such decision. Indeed, let us imagine that $\frac{T_{41}}{\tau_i}$ similarity parameter is changing in such way that it becomes a dominant similarity parameter relative to $\frac{T_{41}}{\tau_g}, \frac{T_{41}}{\tau_d}$. It means that equations (1d) and (1g) become identical, system (1) becomes linear and independent of $V_0$, and a period-doubling bifurcation is impossible:

$$V_{100} = \Phi(\frac{T_{41}}{\tau_i}, I_{41}, I_{23}, I_{22}, I_{19}) \hspace{4cm} \text{(R3)}$$

Thus $\frac{T_{41}}{\tau_i}$ similarity parameter can control what the authors call the middle-Pleistocene transition, and the period-doubling bifurcation can be produced by slow change of $\frac{T_{41}}{\tau_i}$ similarity parameter from its relatively high values to its relatively low values under constant threshold $V_0$. The amplitude of the system response $V_{100}$ will evolve from solution (R3) to solution (R2). Obviously, slow changes of $\frac{T_{41}}{\tau_g}, \frac{T_{41}}{\tau_d}$ may also produce a period-doubling bifurcation. Furthermore, the system (1) has one more "hidden" parameter, i.e., "1" in the glaciation equation. In fact, it is terrestrial ice mass influx that was tacitly set to be constant and equal 1. Recognition of this parameter is important for the exactly same reasons we outlined above for $\tau_i, \tau_g, \tau_d$, and, indeed, it is yet another potential source of bifurcation.

We do not know what specific bifurcation (or their mix) we observe in the historical records and therefore the following question needs to be answered: How sensitive are ***all*** bifurcations, which system (1) may reveal, to our choice of insolation forcing?

Without answering this question the study is incomplete and inconclusive.

Minor comments:

1. In system (1) the glaciation equation is marked as (d) and the deglaciation equation is marked as (g). All references in the text to (g) and (d) states are therefore incorrect.
2. Line 81 "there is no contribution from the obliquity…" It is incorrect, since the obliquity is definitely present in all forcing spectra
3. Line 93 "The importance of orbital forcing alone seems able to start a glaciation…" "The importance" cannot start anything. The phrase needs to be re-formulated.
4. Line 94 "Therefore, the condition to switch from the deglaciation state to the glaciation state is based on insolation only: it is possible to enter ***deglaciation*** when the insolation becomes low enough". You mean "glaciation" here.
5. Line 245 "frow"
6. Line 273 "To model future natural evolutions of the climate system, one would need to take into account for possible evolutions of the $V_0$ threshold." English should be revisited.

References

Verbitsky, M. Y., Crucifix, M., and Volobuev, D. M.: A theory of Pleistocene glacial rhythmicity, Earth Syst. Dynam., 9, 1025–1043, https://doi.org/10.5194/esd-9-1025-2018, 2018.

Verbitsky, M. Y. and Crucifix, M.: π-theorem generalization of the ice-age theory, Earth Syst. Dynam., 11, 281–289, https://doi.org/10.5194/esd-11-281-2020, 2020.

Verbitsky, M. Y. and Crucifix, M.: ESD Ideas: The Peclet number is a cornerstone of the orbital and millennial Pleistocene variability, Earth Syst. Dynam., 12, 63–67, https://doi.org/10.5194/esd-12-63-2021, 2021.

---

## Referee Comment (RC3)

**Additional comment to the manuscript cp-2021-119 "Influence of the choice of insolation forcing on the results of a conceptual glacial cycle model" by Gaëlle Leloup and Didier Paillard**

Dear Andrey,

Certainly, the "gravity acceleration, Plank constant or Milankovitch frequencies" are not tunable parameters, but since they are dimensional, they may be part of adimensional similarity parameters that are indeed tunable. For example, the parameter "viscosity" may not be tunable itself but it is part of the Reynolds number that may change. Therefore, when we create an inventory of governing parameters, physical constants must be included.

I do not think that the number of parameters is a matter of taste because it defines plausibility of the entire model. I have briefly mentioned in my original comment a hidden parameter "1" but may be it deserves a more close consideration.

I hope you and the authors would agree with me that any model of a physical phenomenon should be derivable from the basic laws of physics, providing some assumptions and bearing the "cost" of such assumptions as tunable model parameters. From this perspective, the presented by the authors model (1) is, indeed, an ice sheet mass balance, that is:

$$\frac{dV}{dt} = AS \tag{R21}$$

Here $V$ is ice volume, $S$ is ice sheet area, and $A$ is accumulation minus ablation. All variables here are dimensional. Simply speaking, the changes in ice volume are caused by net accumulation over its entire area. Since $S=V/H$ ($H$ is ice thickness) we can re-write the mass balance as:

$$\frac{dV}{dt} = \frac{V}{\tau} \tag{R22}$$

where $\tau = H/A$.

The ice thickness $H$ is, generally speaking, the function of ice volume, but since $H \sim V^{1/5}$, setting it to be constant may be (reluctantly) accepted. Setting $A$ to be constant is a strong assumption either. The "cost" of these two assumptions are constant timescales adopted by the authors.

The $\frac{V}{\tau}$ term in (R22) is important for ice-sheet dynamics. During the glaciation stage, for example, it is responsible for a positive feedback, specifically: a growing ice sheet spreads as a viscous media increasing its footprint and thus collecting accumulation from a larger area. Indeed, the replacement of $\frac{V}{\tau}$ by $\frac{1}{\tau}$ in the mass balance equation (R22) would be equivalent to changing ice dynamics from $V \sim e^{\frac{t}{\tau}}$ to $V \sim \frac{t}{\tau}$. It may be acceptable on short timescales ($\frac{t}{\tau} < 1$) but on the timescales used in the study ($\frac{t}{\tau} > 1$) the mutation of $\frac{V}{\tau}$ into $\frac{1}{\tau}$ in the glaciation equation (1) needs to have a physical explanation. Formally, in the presented model, the glaciation equation currently contains not just a "hidden" parameter but a "hidden" function $\frac{1}{V}$. This function needs to be exposed and physically described. Without such justification, model (1) cannot be recognized as a physical model and any results may have a somewhat limited

explanatory value. Whatever physical phenomenon is going to be invoked for $\frac{1}{V}$ validation, the "cost" of it will be at least one more governing parameter.

Even if this new parameter (let us for the specificity call it $\lambda$) appears in a ratio $\frac{\lambda}{\tau_g}$ and incomplete similarity in parameters $\lambda$, $\tau_g$ can be claimed, such that $\frac{\lambda}{\tau_g} = \frac{1}{\tau_{g\_new}}$, ($\tau_{g\_new} = \frac{\tau_g}{\lambda}$) the bifurcation trajectory due to the evolution of $\tau_{g\_new}$ can be caused by two physically distinct processes, i.e., by changed ice dynamics ($\tau_g$) and by changing $\lambda$ – physics, whatever the authors designate it to be.

Mikhail Verbitsky

---

## Author Comment (AC1)

Answer to RC 1 :

We thank the reviewer for his helpful comments and suggestions that will help us revise and improve the manuscript. We hope the answers and modifications proposed satisfactorily address his remarks.

In the following, the reviewer's comments are in black, our answer in blue and suggested corrections in green.

G. Leloup and D. Paillard

**Review of the manuscript cp-2021-119 "Influence of the choice of insolation forcing on the results of a conceptual glacial cycle model" by Gaëlle Leloup and Didier Paillard**

The motivation for the research is well articulated. Indeed, the insolation amplitudes under different configurations of the orbital forcing may vary significantly enough to potentially influence our understanding of the climate history. To demonstrate the sensitivity of modeling results to the orbital forcing configuration, the authors offer a number of numerical experiments with a specific dynamical model.

We thank the reviewer for his positive appreciation of our objectives. Still, we want to stress that we are not investigating "different configurations of the orbital forcing" but different choices for the definition of "summer insolation". While the orbital forcing is well-prescribed by celestial mechanics, many different definitions of "summer" have been put forward in the context of the astronomical theory of Quaternary climates. These different choices are usually difficult to justify in the context of conceptual models and our objective is to investigate the robustness of our threshold-based model against these choices.

Unfortunately, the experimental set-up is not comprehensive enough and therefore the results are not conclusive.

The authors are a bit overoptimistic when they introduce their model as the one having only 5 governing parameters. In fact, in the presented set-up, the model has 13 governing parameters, specifically: $I_0$, $V_0$, $\tau_i$, $\tau_g$, $\tau_d$ , $I_{41}$, $I_{23}$, $I_{22}$, $I_{19}$, $T_{41}$, $T_{23}$, $T_{22}$, $T_{19}$

Here $T_{41}$, $T_{23}$, $T_{22}$, $T_{19}$ are orbital periods (an index shows a numerical value in kyr) and $I_{41}$, $I_{23}$, $I_{22}$, $I_{19}$ are corresponding insolation amplitudes. Only timescales and orbital periods have dimension of time, other parameters are adimensional.

If we select, let say, the obliquity period as a parameter with independent dimension, then, according to π-theorem, the amplitude of the 100-kyr system response $V_{100}$ will be a function of 12 adimensional similarity parameters:

$V_{100} = \Phi(I_0,\ V_0,\ T_{41}/\tau_i,\ T_{41}/\tau_g,\ T_{41}/\tau_d,\ I_{41},\ I_{23},\ I_{22},\ I_{19},\ T_{41}/\ T_{23},\ T_{41}/\ T_{22},\ T_{41}/\ T_{19})$ (R1)

It would be physically reasonable to assume $T_{41}/\ T_{23}$, $T_{41}/\ T_{22}$, $T_{41}/\ T_{19}$, and $I_0$ to be constant and, using generalized π-theorem, to migrate to 8 similarity parameters:

$V_{100} = \Phi(V_0,\ T_{41}/\tau_i,\ T_{41}/\tau_g, T_{41}/\tau_d,\ I_{41},\ I_{23},\ I_{22},\ I_{19})$ (R2)

We thank the reviewer for suggesting the use of dimensional analysis and the π - theorem in our study. However, we disagree on the amount of parameters considered. Indeed, the orbital periods are fixed by celestial mechanics (Laskar et al, 2011), and therefore the orbital periods $T_{41}, T_{23}, T_{22}, T_{19}$ cannot be considered as model parameters.

In the same manner, the corresponding insolation amplitudes $I_{41}, I_{23}, I_{22}, I_{19}$, depend on how we define the input summer forcing. For a given input forcing, these amplitudes are fixed. Furthemore, the input forcing (like the summer solstice for example), is not a linear sum of the different obliquity and precession components, and cannot be fully represented by four numbers like $I_{41}, I_{23}, I_{22}, I_{19}$.

We could have chosen for this study multiple possible input forcings with multiple combinations of amplitude of the obliquity or precession periods. However, in this manuscript, we choose to study the model's behavior for specifically four insolation forcings that have been used previously in the litterature as "the Milankovitch forcing" : the summer solstice insolation, the caloric season and the Integrated Summer Insolation (with the use of two different thresholds). Many other choices of "summer insolation" are certainly possible and are not restricted to a 2-parameter (precession, obliquity) or a 4-parameter ($I_{41}, I_{23}, I_{22}, I_{19}$) or a 8-parameter ($I_{41}, I_{23}, I_{22}, I_{19}, \phi_{41}, \phi_{23}, \phi_{22}, \phi_{19}$) space if φ is the phase. The key parameters involved in the definition of "summer insolation" are more likely to be the latitude (see response to RC4), the length of "summer" (caloric = 6 months, solstice = 1 day, above threshold = insolation dependent, …): they are involving some implicit modeling assumptions on the link between astronomy and the dynamics of ice-sheets.

The choice of the input insolation forcing is therefore not a parameter adjustment, but a modeling choice. In answer to a comment of RC2, we have proposed additional discussion of these modeling choices, with the following additions l. 26.

 "In contrast, Milankovitch popularized the idea that the decisive element for glaciation was the presence of cold summers (Berger, 2021), due to reduced summer insolation, at latitudes typical of Northern Hemisphere ice sheets, 65° N. For conceptual models, this raises the question of which insolation to use as input. When summer insolation is used, this questions the definition of summer :  should it be defined as a specific single day, like the summer solstice; the astronomical summer between the two equinoxes; or a fixed number of days around the solstice. This choice leads to very different forcings with different contributions from obliquity and precession. For ESMs and climate models, insolation is computed at each timestep for each grid area, and such choice of the input forcing is not needed. However, other modeling choices have to be made. For instance, several parameterizations are used to represent ice sheet surface mass balance (Robinson et al. 2010), like the Positive Degree Day (PDD) method (Reeh 1991), in which surface melt depends solely on air temperature, or the Insolation Temperature Melt (ITM) method (van den Berg et al., 2008), which takes into account the effect of both temperature and insolation. In both cases, the translation of insolation local and seasonal variations into ice sheet changes and ice age cycles remains an open modeling question."

The authors demonstrated that period-doubling bifurcation caused by rising deglaciation threshold $V_0$ is not very sensitive to the choice of $I_{41}, I_{23}, I_{22}, I_{19}$. Unfortunately, they stopped

here, and, by avoiding variations of $T_{41}/\tau_i, T_{41}/\tau_g, T_{41}/\tau_d$ similarity parameters, they left readers with the impression that changing deglaciation threshold $V_0$ is the only option to reproduce the middle-Pleistocene transition.

In the manuscript, we do not aim at studying all the parameter evolutions that can lead to the MPT (change of a dominated 41 kyr record to a dominated 100 kyr record).
Our purpose is to study the evolution of the deglaciation threshold $V_0$ while other parameters are kept constant, and to see if it is possible to reproduce the geological record by varying solely this parameter.
We decided to focus on this parameter, as it has been identified as crucial by other previous studies (Parrenin and Paillard, 2003; Tzedakis et al 2017).
However, we cannot and do not exclude the fact that the MPT might be obtained by varying other model parameters. The study of all possible parameter modifications leading to the MPT is however out of the scope of this manuscript.
We fully agree that this point was not clearly stated in the first version of the manuscript. This was also noted by RC4 and we therefore propose significant modifications of the manuscript (this is the same answer as to the first general comment of RC4).

We propose the following modification of the abstract (l. 7) :

"Here, we use a simple conceptual model to test and discuss the influence of the use of different summer insolation forcings, having different contributions from precession and obliquity, on the model results. We show that some features are robust. Specifically, to be able to reproduce the frequency shift over the Mid Pleistocene Transition, while having all other model parameters fixed, the deglaciation threshold needs to increase over time, independently of the summer insolation used as input."

We propose the following modification of the end of the introduction (l.61) :

"In particular, we are able to reproduce a switch from 41 kyr oscillations before the MPT to 100 kyr cycles afterwards in agreement with the record for all insolation forcings, by varying a single parameter, the deglaciation threshold $V_0$, and keeping all the other model parameters constant."

We propose the following modification of the conclusion (l.291)

"More specifically, we are able to represent the Mid Pleistocene Transition and the switch from a 41 kyr dominated record to a 100 kyr dominated record, by raising the deglaciation threshold and keeping the other model parameters constant."

l.169, we propose to replace the sentence by "We suggest to replace the sentence by : "In order to study the evolution of the optimal deglaciation threshold $V_0$ over the Quaternary, it was divided into five 500 kyr periods."

l.274, we suggest to add an additional sentence :

 "To model future natural evolutions of the climate system, possible evolutions of the $V_0$ threshold should be considered. However, we do not exclude the fact that variations of other

parameters, that were kept constant in this study, could vary in the future. For instance, different $I_0$ thresholds have to be considered."

As Verbitsky et al (2018) and Verbitsky and Crucifix (2020, 2021) have demonstrated, the space of possibilities to produce a period-doubling bifurcation similar to the middle-Pleistocene transition is much wider and (if we continue to speak in terms of the model being reviewed) $T_{41}/\tau_i$, $T_{41}/\tau_g$, $T_{41}/\tau_d$ similarity parameters definitely have their roles in the drama. The authors' reasoning to keep $\tau_i$, $\tau_g$, $\tau_d$ constant because "these parameters gave correct behaviour in previous studies" is surprising – actually, their past experience with $\tau_i$, $\tau_g$, $\tau_d$ speaks about importance of these parameters for the system dynamics and therefore strongly advocates for them to be in the center of the study. Formally, making $\tau_i$, $\tau_g$, $\tau_d$ constant is equivalent to claiming complete similarity in parameters $T_{41}/\tau_i$, $T_{41}/\tau_g$, $T_{41}/\tau_d$ and excluding them from the solution (R2). Since $T_{41}/\tau_i$, $T_{41}/\tau_g$, $T_{41}/\tau_d$ reference values are of the same order of magnitude as $V_0$, there is absolutely no physical justification for such decision. Indeed, let us imagine that $T_{41}/\tau_i$ similarity parameter is changing in such way that it becomes a dominant similarity parameter relative to $T_{41}/\tau_g$, $T_{41}/\tau_d$. It means that equations (1d) and (1g) become identical, system (1) becomes linear and independent of $V_0$, and a period-doubling bifurcation is impossible:

$V_{100} = \Phi(T_{41}/\tau_i,\ I_{41},\ I_{23},\ I_{22},\ I_{19})$ (R3)

Thus $T_{41}/\tau_i$ similarity parameter can control what the authors call the middle-Pleistocene transition, and the period-doubling bifurcation can be produced by slow change of $T_{41}/\tau_i$ similarity parameter from its relatively high values to its relatively low values under constant threshold $V_0$. The amplitude of the system response $V_{100}$ will evolve from solution (R3) to solution (R2). Obviously, slow changes of $T_{41}/\tau_g$, $T_{41}/\tau_d$ may also produce a period-doubling bifurcation.

In our model, the MPT is not linked to a period-doubling mechanism, but to a frequency- or phase-locking to various astronomical periodicities. Still, we thank the reviewer for mentioning the studies of Verbitsky et al (2018) and Verbitsky and Crucifix (2020, 2021), and we suggest to add a reference to the Verbitsky et al (2018) study in the introduction, l.39.
The new sentence reads :
"The 100 kyr cycles have been proposed to be linked to either eccentricity driven variations of precession (Raymo, 1997; Lisiecki, 2020), obliquity (Huybers and Wunsch, 2005; Liu et al, 2008), or both (Huybers, 2011; Parrenin and Paillard, 2012), to internal oscillations phase locked to the astronomical forcing (Saltzman et al., 1984; Paillard, 1998; Gildor and Tziperman, 2000; Tziperman et al., 2006), to internal oscillations independent of the astronomical forcing (Saltzman and Sutera, 1987; Toggweiler, 2008) or to period doubling bifurcation (Verbitsky et al 2018).

It would be certainly valuable to study the model's behavior to changes of all parameters. But as mentioned above, even the complete list of "parameters" involved in the very definition of summer is subject to lengthy discussions. In any case, this is not the focus of our manuscript.

Furthermore, the system (1) has one more "hidden" parameter, i.e., "1" in the glaciation equation. In fact, it is terrestrial ice mass influx that was tacitly set to be constant and equal

1. Recognition of this parameter is important for the exactly same reasons we outlined above for $\tau_i$, $\tau_g$, $\tau_d$ , and, indeed, it is yet another potential source of bifurcation.
We refer the reviewer to the answer to his comment RC3 for a discussion of this question.

We do not know what specific bifurcation (or their mix) we observe in the historical records and therefore the following question needs to be answered: How sensitive are all bifurcations, which system (1) may reveal, to our choice of insolation forcing?
Without answering this question the study is incomplete and inconclusive.

Our goal is not to study the answer of this kind of model to all possible parameter changes. Our purpose is to study the model's behavior under different definitions of the "summer insolation forcing", and to examine its influence on the evolution of the deglaciation threshold $V_0$, while having other parameters fixed. To our knowledge, it has never been done in the past to force this kind of conceptual model with different kind of insolation forcings, and to compare the induced changes in the results.

Minor comments:
1. In system (1) the glaciation equation is marked as (d) and the deglaciation equation is marked as (g). All references in the text to (g) and (d) states are therefore incorrect.
Indeed, the (d) and (g) labels have been inverted in equation (1). This will be corrected in the next version of the manuscript.

2. Line 81 "there is no contribution from the obliquity..." It is incorrect, since the obliquity is definitely present in all forcing spectra
Indeed, our formulation is confusing. Compared to the work of Parrenin and Paillard (2003), there is no explicit "obliquity term" in the equations. For instance, in Parrenin and Paillard (2003), the glaciation equation reads : $dv/dt = - I_{tr} / \tau_i - O / \tau_O + 1/ \tau_g$ , with O being the obliquity.  In our model, the "obliquity term" $O / \tau_O$ has been deleted. However, as the reviewer stated, there are contributions from the obliquity in the input insolation forcing, as is shown by the spectral analysis of Figure 1 of the manuscript.
We propose to remove the sentence l. 80 to avoid this confusion.

3. Line 93 "The importance of orbital forcing alone seems able to start a glaciation..." "The importance" cannot start anything. The phrase needs to be re-formulated.
We propose to rephrase the sentence and to invert the order of the paragraph for clarity. This gives :

"A critical point is to define the criteria for the switch between the glaciation and deglaciation states. To enter the deglaciation state, both ice volume and insolation seem to play a role (Raymo, 1997; Parrenin and Paillard, 2003, 2012), as terminations occur after considerable build-up of ice sheet over the last million year. To represent the role of both ice volume and insolation in the triggering of deglaciations, the condition to switch from (g) to (d) state uses a linear combination of ice volume and insolation. The deglaciation is triggered when the combination crosses a defined threshold $V_0$ : the deglaciation threshold. As in the work of Parrenin and Paillard (2003), this allows transitions to occur with moderate insolation when the ice volume is large enough and reciprocally. On the contrary, glacial inceptions seem to depend on orbital forcing alone (Khodri et al, 2001; Ganopolski and Calov, 2011). Therefore,

the condition to switch from the deglaciation state to the glaciation state is based on insolation only : it is possible to enter deglaciation when the insolation becomes low enough.”

4. Line 94 “Therefore, the condition to switch from the deglaciation state to the glaciation state is based on insolation only: it is possible to enter deglaciation when the insolation becomes low enough”. You mean “glaciation” here.
Indeed, this will be corrected in the next version of the manuscript.

5. Line 245 “frow”
Indeed, this will be corrected in the next version of the manuscript.

6. Line 273 “To model future natural evolutions of the climate system, one would need to take into account for
possible evolutions of the $V_0$ threshold.” English should be revisited.
We suggest rephrasing by :  “To model future natural evolutions of the climate system, possible evolutions of the $V_0$ threshold should be considered.”

References :
J. Laskar, A. et al (2011). “La2010: a new orbital solution for the long-term motion of the Earth”, Astronomy and Astrophysics, Volume 532,
https://doi.org/10.1051/0004-6361/201116836

F. Parrenin, and D. Paillard (2003). “Amplitude and phase of glacial cycles from a conceptual model”. Earth and Planetary Science Letters. 214. 243-250. 10.1016/S0012-821X(03)00363-7.

P.C. Tzedakis, et al (2017). “A simple rule to determine which insolation cycles lead to interglacials.” *Nature* **542,** 427–432 (2017). https://doi.org/10.1038/nature21364

Reeh, N. (1991) : Parameterization of Melt Rate and Surface Temperature in the Greenland Ice Sheet, Polarforschung, Bremerhaven, Alfred Wegener Institute for Polar and Marine Research & German Society of Polar Research, 59(3), pp. 113-128.

Robinson, A., et al (2010) : An efficient regional energy-moisture balance model for simulation of the Greenland Ice Sheet response to climate change, The Cryosphere, 4, 129–144, https://doi.org/10.5194/tc-4-129-2010, 2010.

van den Berg, J., et al (2008) : A mass balance model for the Eurasian Ice Sheet for the last 120,000 years, Global Planet. Change, 61, 194–208, https://doi.org/10.1016/j.gloplacha.2007.08.015, 2008.

---

## Author Comment (AC2)

Answer to RC 3 :

We thank the reviewer for his helpful comments and suggestions that will help us revise and improve the manuscript. We hope the answers and modifications proposed satisfactorily address his remarks.

In the following, the reviewer's comments are in black, our answer in blue and suggested corrections in green.

G. Leloup and D. Paillard

**Additional comment to the manuscript cp-2021-119 "Influence of the choice of insolation forcing on the results of a conceptual glacial cycle model" by Gaëlle Leloup and Didier Paillard**

Dear Andrey,

Certainly, the "gravity acceleration, Plank constant or Milankovitch frequencies" are not tunable parameters, but since they are dimensional, they may be part of adimensional similarity parameters that are indeed tunable. For example, the parameter "viscosity" may not be tunable itself but it is part of the Reynolds number that may change. Therefore, when we create an inventory of governing parameters, physical constants must be included.

I do not think that the number of parameters is a matter of taste because it defines plausibility of the entire model. I have briefly mentioned in my original comment a hidden parameter "1" but may be it deserves a more close consideration.

I hope you and the authors would agree with me that any model of a physical phenomenon should be derivable from the basic laws of physics, providing some assumptions and bearing the "cost" of such assumptions as tunable model parameters.

We have to disagree with this statement. Different kinds of models, with different kinds of assumptions, allow to study different kinds of questions. Our approach differs from the one of Verbitsky et al (2018). Our model is not physically based, but is a phenomenological model. It certainly does not pretend to be "derived from" or "based on" physical laws, but only pretend to be "consistent" with physics. Our assumptions are therefore not "tunable parameters" but, more simply, modeling choices that are convenient to economically reproduce the phenomenology of ice ages.

Ice sheet mass change is driven by various processes, affecting surface mass balance, ice discharge to the ocean and bottom melt of grounded ice. Here, we do not intend to explicitly represent the numerous physical processes involved in ice sheet volume evolution. The aim of our conceptual model is not to explicitly represent physical processes but rather to help us understand some critical aspects of the climate system.

Our non-physical model can however help us to raise physical questions. To fit the geological record with our model, the deglaciation threshold needs to increase when other parameters are kept constant. This kind of model raises the question of what physical phenomena could be responsible for making deglaciations "harder" to start on the latest part of the Quaternary compared to the earliest part (Paillard, 1998; Tzedakis et al 2017). We demonstrate here that this conclusion does not depend on insolation choices. However, this question cannot be answered with our model.

We propose to clarify the use of conceptual models in the manuscript with addition to the conclusion, l. 296.

"More generally, this kind of glacial-interglacial conceptual model is designed to explain the main features of the Quaternary time period characterized by the waning and waxing of Northern Hemisphere ice sheets under the influence of changing astronomical parameters. In our case, this raises the question of what physical phenomena are responsible for making deglaciations "harder" to start on the latest part of the Quaternary compared to the earliest part. This kind of model is however unlikely to be directly applicable in a more general context, like the Pliocene and earlier periods, or in the context of future climates under the long-term persistence of anthropogenic $CO_2$ (Archer and Ganopolski, 2005; Talento and Ganopolski, 2021)."

Modifications of the description of the conceptual model are proposed afterwards.

From this perspective, the presented by the authors model (1) is, indeed, an ice sheet mass balance, that is:

$dV/dt = AS$ (R21)

Here $V$ is ice volume, S is ice sheet area, and A is accumulation minus ablation. All variables here are dimensional. Simply speaking, the changes in ice volume are caused by net accumulation over its entire area.

Since S=V/H (H is ice thickness) we can re-write the mass balance as:

$dV/dt = V/\tau$ (R22)

where $\tau = H/A$.

The ice thickness H is, generally speaking, the function of ice volume, but since $H \sim V^{1/5}$, setting it to be constant may be (reluctantly) accepted. Setting A to be constant is a strong assumption either. The "cost" of these two assumptions are constant timescales adopted by the authors.

The $V/\tau$ term in (R22) is important for ice-sheet dynamics. During the glaciation stage, for example, it is responsible for a positive feedback, specifically: a growing ice sheet spreads as a viscous media increasing its footprint and thus collecting accumulation from a larger area. Indeed, the replacement of $V/\tau$ by $1/\tau$ in the mass balance equation (R22) would be equivalent to changing ice dynamics from $V \sim e^{t/\tau}$ to $V \sim t/\tau$ . It may be acceptable on short timescales ( $t/\tau < 1$ ) but on the timescales used in the study ( $t/\tau > 1$ ) the mutation of $V/\tau$ into $1/\tau$ in the glaciation equation (1) needs to have a physical explanation. Formally, in the presented model, the glaciation equation currently contains not just a "hidden" parameter but a "hidden" function $1/V$ . This function needs to be exposed and physically described. Without such justification, model (1) cannot be recognized as a physical model and any results may have a somewhat limited explanatory value. Whatever physical phenomenon is going to be invoked for $1/V$ validation, the "cost" of it will be at least one more governing parameter.

Indeed, we agree with the reviewer that our model is not a physical model, We therefore do not provide any explicit physical explanation to "justify" the precise formulation of our equations.
The terms $V/\tau_d$ and $1/\tau_g$ represent trends linked to the current state of the system : slow glaciations and quick deglaciations.

The term $1/\tau_g$ allows to account for processes that do not depend on the ice sheet area and may represent a function of many possible physical phenomena (ice sheet basal temperature for instance is certainly a key "long-term" physical variable, but many other candidates are likely to be involved - isostasy, carbon cycle, to name a few…)

We suggest some additions (starting l.70) to the manuscript to make this point clearer :

"For the glacial-interglacial cycles, it is not a new idea that the climate system can be represented by relaxation oscillations between multiple equilibria, like a glaciation and a deglaciation state (Paillard, 1998; Parrenin and Paillard, 2003, 2012).
The model used in our study is an adapted and simplified version of the conceptual model of (Parrenin and Paillard, 2003). The aim of conceptual models is not to explicitly model and represent physical processes but rather to help us understand critical aspects of the climate system. Here, we do not intend to explicitly represent the numerous physical processes involved in ice sheet volume evolution, affecting surface mass balance, ice discharge to the ocean and bottom melt of grounded ice. Instead, we represent the climate system by two distinct states of evolution : the "glaciation state" (g) and "deglaciation state" (d).
We make the assumption that the evolution of the ice sheet volume in these two states can be simply described by two terms. The first one, common to the glaciation and deglaciation states, is a linear relation to the summer insolation : when the insolation is above average, the ice sheet melts, whereas when the insolation is low enough, the ice sheet grows. The second term, specific to the system state, represents an evolution trend linked to the system state : a slow glaciation in (g) state and a rapid deglaciation in (d) state.

The evolution of the ice volume in these two states in our model is described by :

(g) dv/dt = -I / $\tau_i$ + 1/ $\tau_g$
(d) dv/dt = -I / $\tau_i$ - V/ $\tau_d$

where $v$ represents the normalized ice volume. $\tau_i$, $\tau_d$ $\tau_g$, are time constants.
$I$ is the normalized summer insolation forcing at 65° N, a typical latitude for Northern Hemisphere ice sheets."

Even if this new parameter (let us for the specificity call it $\lambda$) appears in a ratio $\lambda/\tau g$ and incomplete similarity in parameters $\lambda$, $\tau g$ can be claimed, such that $\lambda/\tau g = 1/\tau g\_new$, ($\tau g\_new = \tau g/\lambda$ ) the bifurcation trajectory due to the evolution of $\tau g\_new$ can be caused by two physically distinct processes, i.e., by changed ice dynamics ($\tau g$) and by changing $\lambda$ – physics, whatever the authors designate it to be.

The "1" is not a hidden parameter, as we could rewrite the equation (1g) :
dV/ dt = -I / $\tau_i$ + $\alpha$ with $\alpha =$ 1/$\tau_g$.
The $\alpha =$ 1/$\tau_g$ parameter could be linked to various physical processes, that we do not intend to represent explicitly with our model.
A complete parameter enumeration of our conceptual model (which is not physically based) is therefore a list of 5 mathematical parameters ($I_0$, $V_0$, $\tau_i$, $\tau_g$, $\tau_d$) while the true physics behind may likely involve many more physical ones.

---

## Author Comment (AC3)

Answer to RC2 :

We thank the reviewer for his helpful comments and suggestions that will help us revise and improve the manuscript. We hope the answers and modifications proposed satisfactorily address his remarks.

In the following, the reviewer's comments are in black, our answer in blue and suggested corrections in green.

G. Leloup and D. Paillard

The manuscript by Leloup and Paillard tests the ability of a simple conceptual model to simulate Quaternary glacial cycles using different metrics of "orbital forcing" which differ by relative contributions of precessional and obliquity components. The authors used only one parameter (critical ice volume) to maximize model performance in term of a novel performance metric proposed by the authors. The main results of this study can be summarized as follows:

i) model performance does not strongly depend on whether "orbital forcing" is dominated by precession or precession is essentially absent.

ii) the transition from short to long glacial cycles in all cases can be achieved by an increase of the critical ice volume by factor 2-2.5.

While the latter is not surprising since 100/40=2.5, the first result requires more serious discussion (see below).

General comments

1. Since the review of Mikhail Verbitsky was already available at the time when I was requested to write my own, it is natural that I read his review before writing. And I must respectfully disagree with Mikhail in respect of the number of model parameters used in this study. Under "model parameters" (at least in our field) we understand parameters that can be used for model tuning. Gravity acceleration, Plank constant or Milankovitch' frequencies are not such parameters. This is why, formally, the Leloup and Paillard model (which is a simplified version of Parennin and Paillard mode) has only 5 parameters. In fact, the authors did not use four of them for model tuning since they used values for these parameters from a different model (Parennin and Paillard, 2003). Thus, the only parameter which the authors used for model tuning is the critical ice volume. Whether it is good on not is another issue.

We fully agree with the reviewer's comment and thank him for taking the time to read previously published comments. We refer the reviewer to our answers to RC1 and RC3 for more discussion.

2. When authors discuss the current state of our understanding of Quaternary climate variability, they are too pessimistic. The authors repeat twice (in abstract and introduction) that "the nature and physics of the [link between insolation and the glacial - interglacial cycles] remain unclear", and "the Mid Pleistocene transition ... remain mostly unexplained". Such statements would be, probably, appropriate in 1998 but not in 2021. Of course, some questions remain and, likely, will remain for some time but the major issues are already clear.

Indeed, we do understand now much more than 20 years ago. We suggest to rephrase the introduction in the following way (l.21) : "[...] suggesting a strong link between insolation and the glacial - interglacial cycles. The nature and physics of this link has been a central question since the discovery of previous warm and cold periods [...]"

However, discussions remain open on modeling choices of the link between insolation and ice sheet evolution. This is discussed later on (answer to the specific comment relative to l.26).

3. Of course, it is up to the authors to decide which model to employ, which parameters use to tune the model and which criteria use to select the optimal parameters set. However, as the result of authors' choice, the model performance for all four "orbital forcings" for post-MPT glacial cycles are essentially the same. The authors claim that "we are able to represent the Mid Pleistocene Transition and the switch from a 41 kyr dominated record to a 100 kyr dominated record, by raising the deglaciation threshold (L. 291)." However, Fig. 4 clearly shows that this is not the case, since for three of four "orbital forcings" obliquity continues to dominate after MPT. Only for the solstice insolation, this is not the case, but then the model instead of sharp 100 kyr cyclicity simulates something which looks more like a red noise. Thus, as far as spectral properties of simulated glacial cycles are concerned, none of the model realisations is really successful. Whether this is a result of model formulation, fixing of four of five model parameters, or criteria for optimization - is not clear to me but has to be discussed in the paper.

We agree with the reviewer that the model data mismatch is not sufficiently discussed in the original version of the manuscript. We propose a rewriting of the corresponding discussion part (replacing l. 231 to 248) with first, a discussion of the timing based on our "accuracy criteria"( are the terminations at the right place?) and second, a discussion of its spectral properties. Indeed, the timing of terminations and the spectral properties are two rather different questions, though they are of course related. In the original manuscript, we focussed mostly on the first question by defining our "accuracy criteria", but the second one is indeed unavoidable.

"Our conceptual model is able to reproduce qualitatively well the data (LR 04 normalized curve) over the whole Quaternary. The model's best fit over the Quaternary for each insolation forcing, as defined in Section 2.3*, is displayed in Fig. 4. It is able to reproduce the frequency shift from a dominant 41 kyr period before 1 Ma BP to longer cycles afterwards, as observed in the data, and thus by varying only one parameter during the whole simulation length : the deglaciation threshold $V_0$.

For every input forcing, longer cycles are obtained on the last part of the Quaternary (last Myr). Figure 4 displays the results over the whole Quaternary with the $V_0$ threshold being set to its optimal value on each 500 kyr period, while Figure 5 displays the results over the last million year with the $V_0$ threshold being set to its optimal value on the [0 - 1000] kyr BP period.

For the last million year, it is possible to reproduce with the right timing all terminations, apart from the last deglaciation, for all insolation forcings, with a single value for the $V_0$ threshold over this period. Some differences are however noticeable between the different forcings. Especially for the ISI above 300 W/m$^2$ forcing, the agreement is not as good as for the other forcings : Termination V (around 420 kyr BP) is triggered later compared to the data, while Termination III (around 240 kyr BP) is triggered too early. For the ISI above 300 W/m$^2$ forcing, the range of $V_0$ values allowing to reproduce correctly most of the terminations on the last million year is reduced (only values of $V_0$ = 3.9 - 4.0), whereas the results are more robust for the three other insolations forcings, with a broader range of working $V_0$ values. The ISI above 300 W/m$^2$ forcing has a low precession component, which explains why it is

less successful in reproducing the data over the last million year. Experiments with our model setup have shown that a summer forcing with no precession component could not successfully reproduce the data over the post MPT period as accurately as the four forcings presented here, that contain both precession and obliquity**.

Despite the accurate timing of terminations, the spectral analysis of the model results over the last million year differs from the spectral analysis of the data. For all forcings except the summer solstice insolation, obliquity continues to dominate after the MPT. The spectral analysis shows secondary and third peaks of lower frequency, but does not show a sharp 100 kyr cyclicity as in the LR04 record. Compared to the data, all the model outputs over the post MPT period have a more pronounced obliquity and precession component and a less pronounced 100 kyr component. This feature is most probably due to the model formulation, and more specifically the direct dependence of ice volume evolution to insolation via the $dV/dt = - I / \tau_i$ term. This is one of the limits of our conceptual model. While the criteria on the switch to deglaciations allows us to reproduce the deglaciations at the right timing, the direct dependance of ice volume change to the insolation forcing is definitely too simplistic and probably produces an overestimated dependency of the ice evolution to the astronomical forcing on the latter part of the record.

On the first part of the Quaternary (2.6 Ma BP to 1 Ma BP), the spectral analysis of the data is dominated by a 41 kyr (obliquity) peak. It is also the case for the model results, for each type of insolation. However, the model outputs also show a precession component (19 to 23 kyr), especially for the summer solstice and the Integrated Summer Insolation above 400 W/m$^2$ forcings, which does not exist on the data.

*The model best fit was not clearly defined in the first version of the manuscript, but will be in Section 2.3 in the next version. We refer to the answer to the reviewer's minor comment concerning this point, or to question 6 of RC4 for modification suggestions.*

**Here we will refer the reader to the answer of question 3 of RC4*

[Figure]

*[Figure 4 of the manuscript, with normalization changed for the spectral analysis (it was normalized by the maximum value in the manuscript, and is now normalized by the standard deviation)]*

[Figure]

*[Figure 5 of the manuscript]*

4. As I already mentioned above, if the authors are convinced that model results are equally realistic irrespectively of whether "orbital forcing" contains a strong precessional component (solstice insolation) or almost none (ISI), then they should conclude that precession plays no role in Quaternary glacial cycles and, thus, 100 kyr cyclicity has nothing to do with eccentricity. Do the authors agree with such a statement? Please comment.

We thank the reviewer for this question, that will allow us to clarify the manuscript. We certainly do think that precession has a key role in the 100 kyr cycles.

Even if the model fit over the last million year for the ISI above 300 W/m² forcing has an accuracy comparable to the other forcings, its agreement with the data is poorer. We hope that the reformulation of the discussion section proposed above clarified this fact.

To answer question 3 of RC 4, we have furthermore performed additional simulations. In these simulations, we compared the model results in a case of a "pure obliquity forcing" : we have computed the summer solstice insolation at 65° N, but with a fixed value for the precession parameter, resulting in a signal that has no precession component.

These results show that it is not possible to reproduce accurately the terminations on the last million year with a fixed $V_0$ with this input forcing having no precession component.

Even if its precession component is low, the ISI above 300 W/m² forcing allows to better represent the post MPT part of the record compared to a forcing without precession, not only in terms of spectral content but especially in terms of terminations timing.

5. I do not understand what is shown in fig 1. Obviously, the figure heading (Normalized summer solstice insolation) is not applicable to the entire figure. More important is that the upper panel does NOT show summer solstice insolation. What it shows - I do not know.

*Indeed, the figure heading will be changed in the next version. Also, the top panel mistakenly shows the summer solstice insolation between 2 and 3 Myr BP, instead of between 0 and 1 Myr BP. There is the same problem with the caloric season on the second panel. This will be corrected in the next version.*

*Here is the new version of Figure 1. The normalization of the spectral analysis has also been changed (normalization by the standard deviation, and not by the maximum value as previously).*

[Figure]

*Figure : [New version of Figure 1 of the manuscript ] (a) The four different summer insolation forcings at 65° N (summer solstice, caloric season, Integrated Summer Insolation above 300 W/m² and above 400 W/m²) displayed over the last million year (respectively in blue, brown, pink and orange). (b) Corresponding spectral analysis.*

Specific comments

L. 24. Milankovitch not just "popularized" this idea (which was not his own idea) but made it the key element of his ice age theory.

We propose the following reformulation based on the reviewer's suggestion.

"In contrast, the idea that the decisive element for glaciation was the presence of cold summers, due to reduced summer insolation, at latitudes typical of Northern Hemisphere ice sheets, 65° N, was taken up by Milankovitch. He made it the key element of his ice age theory (Berger, 2021)"

L. 26. "This also raises the question of what period should be defined as summer". It should be made clear that the question of how to define "summer insolation" is relevant only for conceptual models, like one used in this study. Climate models and ESMs compute insolation at each time step for each grid-cell and do not need such prescriptions.

Indeed, the question of the definition of summer insolation is only relevant for conceptual models. However, we would like to stress the fact that even for ESMs and climate models, the question of the link between insolation and ice sheet changes is still an open modeling question. We therefore suggest to reformulate and to add additional sentences, starting l.26.

"In contrast, the idea that the decisive element for glaciation was the presence of cold summers, due to reduced summer insolation, at latitudes typical of Northern Hemisphere ice sheets, 65° N, was taken up by Milankovitch. He made it the key element of his ice age theory (Berger, 2021). For conceptual models, this raises the question of which insolation to use as input. When summer insolation is used, this questions the definition of summer : should it be defined as a specific single day, like the summer solstice; the astronomical summer between the two equinoxes; or a fixed number of days around the solstice. This choice leads to very different forcings with different contributions from obliquity and precession. For ESMs and climate models, insolation is computed at each timestep for each grid area, and such choice of the input forcing is not needed. However, other modeling choices have to be made. For instance, several parameterizations are used to represent ice sheet surface melt (Robinson et al. 2010), like the Positive Degree Day (PDD) method (Reeh 1991), in which surface melt depends solely on air temperature, or the Insolation Temperature Melt (ITM) method (van den Berg et al., 2008), which takes into account the effect of both temperature and insolation. In both cases, the translation of insolation local and seasonal variations into ice sheet changes and ice age cycles remains an open modeling question."

And to clarify l.48 that the choice of the input insolation is critical for not all, but conceptual models.

This gives (l.48) : "One of the critical questions for conceptual models is to decide which insolation to use as input."

L. 28. This choice leads to very different forcings.

This will be corrected in the next version of the manuscript.

L. 41. The authors should make it very clear that they only consider here conceptual models of glacial cycles.

Indeed, this will be made clearer in the next version. We suggest to rephrase (l.41) by : "Several conceptual models have been developed to try to solve these questions."

L. 81, 93 and 94. I fully agree with the comments by Mikhail Verbitsky

This will be corrected in the next version of the manuscript. Please refer to the answer to RC1 for corresponding modifications.

Eq. 2. Please change V to v.

This will be corrected in the next version of the manuscript.

Last par, p. 7. When discussing pre-MPT model performance, it is important to realise that for this period of time, LR04 stack was tuned to obliquity. This is why it is not surprising that it contains nothing apart from obliquity

Indeed. Furthermore the LR04 record is not a direct representation of ice volume changes. We propose to enhance the different limits to the use of the LR 04 stack as a proxy for ice volume changes, starting l. 208 :

"Moreover, the $\delta^{18}O$ LR04 curve includes at the same time an ice volume and deep water temperature component. Ice volume and sea level reconstructions do exist (Bintanja, 2005; Spratt and Lisiecki, 2016), but are however limited to the more recent part of the Quaternary and do not allow the investigation of the pre MPT period. The use of $\delta^{18}O$ as an ice volume proxy has already been largely debated (Schackleton, 1967; Chappell and Schackleton, 1986; Schackleton and Opdyke, 1973; Clark et al., 2006) and recent studies (Elderfield et al., 2012) have shown that the temperature component may be as large as 50%. Furthemore, the stack was tuned to insolation (Lisiecki and Raymo, 2005). We refer the reader to (Raymo et al 2018) for a review of possible biases in the interpretation of the LR 04 benthic $\delta^{18}O$ stack as an ice volume and sea level reconstruction. All these reasons encourage us to remain at a qualitative level to fit the data."

L. 231. Which "data"? What "best guess" means?

In Section 2.3, l. 146 we introduce the data used, and propose the following reformulation :

 "To compare our model results to data, we used the benthic $\delta^{18}O$ stack "LR04" (Lisiecki and Raymo, 2005) as a proxy for ice volume, considering that most of the $\delta^{18}O$ changes of benthic foraminifera represent changes in continental ice (Schackleton, 1967; Schackleton and Opdyke, 1973)".

We suggest to add an additional sentence, after l.145 : "Lower $\delta^{18}O$ values correspond to lower ice volume. The model results as well as the LR04 curve were normalized to facilitate their comparison. In the following "data" refers to the $\delta^{18}O$ stack curve LR04 normalized."

The use of the LR04 stack curve as data is only stated once in the text of manuscript, and we acknowledge that we should refer to it more often to improve clarity.

We suggest to add an explicit reference to it at the start of the Section 3.2 (l. 231) : "Our conceptual model is able to reproduce qualitatively well the data (normalized LR04 curve) over the whole Quaternary".

We agree with the reviewer that the model "best fit over the Quaternary"/ "best guess" was not clearly defined in the manuscript. This issue was also raised by RC4.

We suggest rephrase the end of section 2.3 (starting l.170) in coherence with the modifications suggested to RC4. In section 3.2, we will refer the reader to the definition given in section 2.3. The reformulation reads :

"The $V_0$ values that maximize the accuracy criteria for each time period and insolation forcing are called 'optimal $V_0$'. To determine the optimal $V_0$ threshold corresponding to each period and insolation forcing, several simulations were carried out and the parameter values maximizing the accuracy criteria $c$ were selected. More precisely, for each insolation and period, 3500 simulations corresponding to different $V_0$ thresholds (from $V_0=1.0$ to $V_0=8.0$ with a step of 0.1) and different initial conditions (initial volume $V_{init}$ ranging from 0.0 to 5.0 with a step of 0.2, and initial state - glaciation or deglaciation) were performed. For each insolation forcing, the best fit over the Quaternary is defined as the simulation over the whole Quaternary (0- 2500 kyr BP) with a $V_0$ changing with time, and that is equal to the corresponding optimal $V_0$ at each time period."

L. 300. Talento and Ganopolski is now published

This will be corrected in the next version of the manuscript.

References :

Reeh, N. (1991) : Parameterization of Melt Rate and Surface Temperature in the Greenland Ice Sheet, Polarforschung, Bremerhaven, Alfred Wegener Institute for Polar and Marine Research & German Society of Polar Research, 59(3), pp. 113-128.

Robinson, A., et al (2010) : An efficient regional energy-moisture balance model for simulation of the Greenland Ice Sheet response to climate change, The Cryosphere, 4, 129–144, https://doi.org/10.5194/tc-4-129-2010, 2010.

van den Berg, J., et al (2008) : A mass balance model for the Eurasian Ice Sheet for the last 120,000 years, Global Planet. Change, 61, 194–208, https://doi.org/10.1016/j.gloplacha.2007.08.015, 2008.

---

## Author Comment (AC4)

Answer to RC4 :

We thank the reviewer for her/his helpful comments and suggestions that will help us revise and improve the manuscript. We hope the answers and modifications proposed satisfactorily address her/his remarks.

In the following, the reviewer's comments are in black, our answer in blue and suggested corrections in green.

The additional figures performed are labelled Qa-b with a the number of the reviewer's question and b the number of the corresponding figure in the original manuscript.

G. Leloup and D. Paillard

**General comments**

1. I think that it should be clearly stated in the abstract, introduction and conclusions that the paper will focus on the deglaciation threshold V0, while keeping all the other 4 model parameters constant. In particular, I find the sentence "to be able to reproduce the frequency shift over the Mid Pleistocene Transition, the deglaciation threshold needs to increase over time, independently of the summer insolation used as input" misleading if it is not pointed out that all the other parameters are fixed. If temporal changes in the other parameters were also allowed, then the frequency shift over the MPT could potentially also be reproduced via other changes.

We fully agree with the reviewer's comment. This will be made clearer in the next version of the manuscript.

We propose the following modification of the abstract (l. 7) :

"Here, we use a simple conceptual model to test and discuss the influence of the use of different summer insolation forcings, having different contributions from precession and obliquity, on the model results. We show that some features are robust. Specifically, to be able to reproduce the frequency shift over the Mid Pleistocene Transition, while having all other model parameters fixed, the deglaciation threshold needs to increase over time, independently of the summer insolation used as input."

We propose the following modification of the end of the introduction (l.61) :

"In particular, we are able to reproduce a switch from 41 kyr oscillations before the MPT to 100 kyr cycles afterwards in agreement with the records for all insolation forcings, by varying a single parameter, the deglaciation threshold $V_0$, and keeping all the other model parameters constant"

We propose the following modification of the conclusion (l.291)

"More specifically, we are able to represent the Mid Pleistocene Transition and the switch from a 41 kyr dominated record to a 100 kyr dominated record, by raising the deglaciation threshold and keeping the other model parameters constant."

2. How do the results change if you consider the insolation at 50°N, as in Caulder's model, instead of 65°N?

This is an interesting question. In his article, Calder used the latitude of 50° N and not 65° N. However, even if we choose the 50° N latitude for our input forcing, our results would hardly be comparable to the one of Calder's model, as the astronomical solutions have changed since. Calder used the Vernekar tables (1972), while we use the Laskar 04 (Laskar, 2004) solution. For the latitude of 50° N, the question of which definition of the summer insolation to choose (insolation at the summer solstice, caloric season or Integrated Summer Insolation above a threshold) would remain, leading to a new set of experiments that is outside of the scope of this paper.

Here, we have performed a new experiment, in the case of the summer solstice insolation at 50° N, and compared it with the results previously obtained with the 4 different summer insolation types at 65° N.

The new figures, corresponding to the one of the original manuscript with the additional experiment (summer solstice insolation at 50° N used as input) are labeled Q2-1 to Q2-5 (following the number of the figure in the original manuscript).

[Figure]

*Figure Q2-1: [Fig. 1 of the manuscript with summer solstice insolation at 50°N]*

*(a) The four different summer insolation types at 65° N and the summer solstice insolation at 50° N, displayed over the Quaternary period. (b) Corresponding spectral analysis, normalized by the standard deviation.*

The summer solstice insolation at 50° N has a lower obliquity component than the summer solstice insolation at 65° N.

[Figure]

*Figure Q2-2*: [Fig. 2 of the manuscript with summer solstice insolation at 50°N]

*Optimal deglaciation threshold $V_0$ over the five different periods for the four different summer insolation forcings at 65° N and the summer solstice insolation at 50° N. When several values of the deglaciation threshold $V_0$ maximize the accuracy criteria, the mean value is plotted and the other possible values are represented with errorbars.*

[Figure]

*Figure Q2-3*: [Fig. 3 of the manuscript with summer solstice insolation at 50°N]

*Accuracy over the five different periods for the four different summer insolation forcings at 65° N and the summer solstice insolation at 50° N.*

Please note that in Figures 2 and 3, the values over the period [1000 - 1500]kyr BP for the four initial summer forcings (at 65° N) have changed compared to the first version of the manuscript, due to an error in the plot. However, this does not change the main results : the optimal deglaciation threshold $V_0$ increases over the Quaternary, and the accuracy is generally higher on the latter part of the record.

With the summer solstice insolation at 50° N, we obtain the same conclusion as in the case of forcings at 65° N. The optimal deglaciation $V_0$ (obtained while keeping all other model parameter constants), increases over the Quaternary, with lower values around 3 at the start of the Quaternary and values around 5 after the MPT.

As in the case of the summer solstice insolation at 65° N, the accuracy values of the results with the summer solstice insolation at 50° N are higher on the later part of the Quaternary. Compared to the solstice insolation at 65° N, the solstice insolation at 50°N always leads to a lower accuracy on the earliest part of the record. This seems logical as the insolation at 50° N has a weaker obliquity component, and the pre MPT period is dominated by obliquity (41 kyr cycles). Following the accuracy criteria defined in the manuscript, their performances on the post MPT part are comparable.

[Figure]

*Figure Q2-4: [Fig. 4 of the manuscript with summer solstice insolation at 50°N]*

*Best model fit over the whole Quaternary and corresponding spectral analysis. The middle panel represents the best fit of the model for the different summer insolation used as input, compared to the data. The data (normalized LR04 curve) are in black. The blue shading represents deglaciation periods in the data and the yellow shading deglaciation periods in the model. This results in a green shading when deglaciations are seen in the model and data at the same time.The left panel represents the spectral analysis of the best fit solution over the last million year. The right panel represents the spectral analysis over the more ancient part of the Quaternary (before 1 Ma BP)*

On Figure Q2-4, the best guess over the Quaternary was obtained (see answer to Q7 for a definition and corresponding modification suggestions), and the spectral analysis over the pre MPT ([1000 - 2500] kyr BP) and post MPT ([0 - 1000] kyr BP) was carried out.

The spectral analysis over the pre MPT period shows that model results with the summer solstice insolation at 50° N as forcing fail to reproduce the dominance of the obliquity cycle that is visible on the data on that part of the record. As for the accuracy value on this period, this is not surprising, as the summer solstice insolation at 50° N  has a low obliquity

component. On the post MPT part, larger cycles are produced, but their frequency do not match the data.

[Figure]

*Figure Q2-5 :* *[Fig. 5 of the manuscript with summer solstice insolation at 50°N]*

*Normalized model results over the last million year, with the four different summer insolation forcings at 65° N and the summer solstice insolation at 50° N . The full line is the best fit computed over the 100 - 1000 ka BP period. The dashed line represents the same solution, but with an increased $V_0$ threshold for the last deglaciation. The data (LR04 stack curve normalized) are in black. The yellow shading represents deglaciation periods in the model (case of the increased $V_0$ threshold) and the blue shading represents deglaciation periods in the data. This results in a green shading when deglaciations are seen in the model and data at the same time*

On the last million year, the model output with the summer solstice insolation at 50° N as forcing reproduces quite well the data. As in the case of the summer solstice insolation at 65° N, there is no need to increase the $V_0$ threshold over the last 100 kyr in order to reproduce the last cycle. This is visible in Figure Q2-5, where the best fit over the [0 - 1000] kyr BP period with a constant $V_0$ threshold on this period is shown in full lines. The dashed lines represent the case where the $V_0$ threshold is increased over the last 100 kyr. For the summer solstice insolation at 65°N and 50°N, the full and dashed curve overlap, as increasing the $V_0$ threshold over the last 100 kyr does not change the results.

In this manuscript, we aim at comparing "classical" insolation types that could be used to force conceptual models at the latitude of 65° N. We could have chosen different latitudes and definitions of summer insolation, and the amount of possible experiments is infinite. Here, we have shown that the main results of the manuscript do not change with the summer solstice insolation at 50° N. Indeed, we are able to produce a frequency shift of the cycles by increasing the $V_0$ threshold over the MPT. As it contains less obliquity, the results with this forcing are poorer on the earliest part of the Quaternary, where the record is dominated by obliquity. For the clarity of the manuscript, we prefer to stick to the four initial forcings used, at the same latitude of 65° N.

We suggest to add a sentence l.124 in the presentation of the input forcings used, and to refer to the answer to RC4.

"Experiments were also conducted for the summer solstice insolation at 50° N instead of 65° N but are not presented here as they do not change the conclusions obtained with the forcings at 65° N."

3. What happens if you decompose the summer solstice insolation into its precession and obliquity components and use each of them as forcing in your model? By the results shown in the manuscript, the ISI above 300W/m2 forcing (which contains almost no precessional signal) still produces what the authors deem a "good fit to data". Therefore, I think that a clean separation into the precessional and obliquity components of, let's say, summer solstice insolation could be a clean experiment to compare with.

We did not use precession and obliquity as direct forcings, since we intend to have a model forced by the "Milankovitch" forcing, that is a forcing based on "summer insolation". Summer insolation has indeed two main components (obliquity and precession) with varying power depending on insolation definition, but there is no "summer insolation" without precession, nor without obliquity.

Still, in the following, we have compared the model results with 3 different forcings :

- the summer solstice insolation at 65°N as in the manuscript
- the "fixed obliquity / precession only" forcing. This forcing was obtained by computing the summer solstice insolation at 65° N but with a constant obliquity value (equal to the current value)
- the "fixed precession / obliquity only" forcing. This forcing was obtained by computing the summer solstice insolation at 65° N but with a constant value of the precession parameter (equal to the current value)

[Figure]

*Figure Q3-1: [Fig. 1 of the manuscript with summer solstice insolation, fixed obliquity and fixed precession forcing]*

*(a) Summer solstice at 65°N, fixed obliquity and fixed precession forcings, displayed over the Quaternary period. (b) Corresponding spectral analysis, normalized by the standard deviation.*

Figure Q3-1 represents these forcings. As expected, the spectral analysis of the "fixed obliquity" forcing has only components from the precession, and conversely the "fixed precession" forcing has only an obliquity component.

[Figure]

*Figure Q3-2: [Fig. 2 of the manuscript with fixed obliquity and fixed precession forcings]*

*Optimal deglaciation threshold $V_0$ over the five different periods for the summer solstice at 65°N, fixed obliquity and fixed precession forcings. When several values of the deglaciation threshold $V_0$ maximize the accuracy criteria, the mean value is plotted and the other possible values are represented with errorbars.*

[Figure]

*Figure Q3-3: [Fig. 3 of the manuscript with fixed obliquity and fixed precession forcings]*

*Accuracy over the five different periods for the summer solstice at 65°N, fixed obliquity and fixed precession forcings.*

Figure Q3-3 shows that the summer solstice insolation allows a better accuracy in reproducing the last part of the record than the fixed obliquity and fixed precession forcings.

The Figure Q3-4 shows the best fit over the whole Quaternary. With a fixed obliquity (precession only) forcing, our model is not able to reproduce the pre-MPT 40 kyr signal, whereas this is possible with the fixed precession (obliquity only) forcing. It is not surprising that an input forcing with no obliquity component does not allow to reproduce the pre MPT, obliquity-dominated record. On the latest part of the Quaternary, the results obtained are less satisfying with the fixed precession and fixed obliquity forcings than with the summer solstice insolation. Concerning the spectral analysis, for the fixed precession (obliquity only) forcing, obliquity continues to dominate after the MPT. For the fixed obliquity (precession only) forcing, there is a 100 kyr peak after the MPT. However, concerning the terminations placement, both of these forcings fail to successfully represent the post MPT part of the record.

[Figure]

*Figure Q3-4: [Fig. 4 of the manuscript with fixed obliquity and fixed precession forcings]*

*Best model fit over the whole Quaternary and corresponding spectral analysis. The middle panel represents the best fit of the model for the different insolation used as input, compared to the data. The data (normalized LR04 curve) is in black. The blue shading represents deglaciation periods in the data and the yellow shading deglaciation periods in the model. This results in a green shading when deglaciations are seen in the model and data at the same time.The left panel represents the spectral analysis of the best fit solution over the last million year. The right panel represents the spectral analysis over the more ancient part of the Quaternary (before 1 Ma BP)*

Figure Q3-5 shows the best fit over the [0 - 1000] kyr BP period. When the optimization is done over the [0 - 1000] kyr period, the respective accuracy over this period is 0.92 for the summer solstice forcing and 0.67 for both the fixed obliquity and fixed precession forcings. With the fixed obliquity, and fixed precession forcing, some terminations are misplaced.

For instance, with the fixed obliquity (precession only) forcing, Termination V (around 420 kyr BP) is triggered too late. With the fixed precession (obliquity only) forcing, Termination VII and Termination IX (around 620 and 790 kyr BP) are misplaced.

[Figure]

*Figure Q3-5 : [Fig. 5 of the manuscript with fixed obliquity and fixed precession forcings]*

*Normalized model results over the last million year, with the summer solstice at 65°N, fixed obliquity and fixed precession forcings. The colored lines are the best fit computed over the 0 - 1000 ka BP period. The data (LR04 stack curve normalized) is in black. The yellow shading represents deglaciation periods in the model and the blue shading represents deglaciation periods in the data. This results in a green shading when deglaciations are seen in the model and data at the same time*

These results show that when an obliquity only or precession only forcing are used, our model is not able to represent the latest part of the Quaternary record as well as when forcings with contributions from both the precession and the obliquity are used.

In the case of the ISI 300 W/m² forcing, the precession component is small, but is not absent, and allows a better fit to data. With our accuracy criteria, the accuracy of this forcing is comparable to the other forcings, but differences are indeed noticeable when looking at the record.

We have proposed modifications (see answer of question 3 of RC2), and among those, to add the sentence :

"Especially for the ISI above 300 W/m² forcing, the agreement is not as good as for the other forcings : Termination V (around 420 kyr BP) is triggered later compared to the data, while Termination III (around 240 kyr BP) is triggered too early. For the ISI above 300 W/m² forcing, the range of $V_0$ values allowing to reproduce correctly most of the terminations on the last million year is reduced (only values of $V_0$ = 3.9 - 4.0), whereas the results are more robust for the three other insolations forcings, with a broader range of working $V_0$ values. The ISI above 300 W/m² forcing has a low precession component, which explains why it is less successful in reproducing the data over the last million year. Experiments with our model setup have shown that a summer forcing with no precession component could not successfully reproduce the data over the post MPT period as well as the four forcings presented here, that contain both precession and obliquity*.

*\* Here we will refer the reader to the answer of question 3 of RC4*

4. Will the results be the same with a different fitting criterion? For example, if instead of the defined c, the criterion for the optimisation is to maximise the correlation between time-series of normalized paleo and modelled ice volume, how do the results change? It is obvious that the correlation measure will penalise the ISI above 300W/m2 forcing, as it clearly cannot reproduce the variability at the 100 kyr frequency.

The figure corresponding to the case where we try to maximize the coefficient correlation instead of our initial accuracy criteria are displayed below, with the initial figures for comparison. Please note that in Figures Q2 and Q3, the values over the period [1000 - 1500]kyr BP have changed compared to the first version of the manuscript, due to an error in the plot. However, this does not change the main results : the optimal deglaciation threshold $V_0$ increases over the Quaternary, and the accuracy is generally higher on the latter part of the record.

[Figure]

[Figure]

*Figure Q4-2 : [Figure 2 of the manuscript and corresponding figure with the correlation coefficient as fitting criteria]*

*Optimal deglaciation threshold $V_0$ over the five different periods for the four different summer insolation forcings at 65° N. The left part displays the result when the initial accuracy criteria of the manuscript is used. The right part displays the result when the correlation coefficient is used as accuracy criteria. When several values of the deglaciation threshold $V_0$ maximize the accuracy criteria, the mean value is plotted and the other possible values are represented with errorbars.*

Changing the accuracy criteria from the initial criteria proposed in the manuscript to the correlation coefficient changes the value of the optimal $V_0$ threshold, as visible in Figure Q4-2. However, this does not change the general tendency : lower values of the $V_0$ threshold allow to better fit the earliest part of the record, while higher values lead to a better fit on the post MPT period. For all insolation forcings (except the summer solstice insolation, detailed below), there is a clear tendency of increase of the $V_0$ threshold over the different time periods.

The correlation coefficient is better on the latest part of the record compared to the pre MPT part for all insolation forcings, and the summer solstice insolation leads to poorer results than all the other forcings on the pre MPT period, as visible in Figure Q4-3.

[Figure]

Figure Q4-3: [Figure 3 of the manuscript and corresponding figure with the correlation coefficient as fitting criteria]

Accuracy over the five different periods for the four different summer insolation forcings at 65° N.  The right part displays the result when the correlation coefficient is used as accuracy criteria.

[Figure]

Figure Q4-4a: [Figure 4 of the manuscript] Best model fit over the whole Quaternary and corresponding spectral analysis when the accuracy criteria is the initial criteria

[Figure]

*Figure Q4-4b: [Figure 4 with the correlation coefficient as fitting criteria]*

*Best model fit over the whole Quaternary and corresponding spectral analysis when the accuracy criteria is the correlation coefficient.*

On Figure Q4-4a, the optimal fit over the Quaternary when the initial criteria is used as the fitting criterion is displayed. Please note that we have changed the normalization of the spectral analysis (it was normalized by the maximum value in the manuscript, and is now normalized by the standard deviation). On Figure Q4-4b, the optimal fit over the Quaternary when the correlation criteria is used as the fitting criterion is displayed. When the caloric season and the ISI above 300 W/m$^2$ and 400 W/m$^2$ are used as input forcing, the spectral analysis over the pre MPT period does not change much. However, for the summer solstice insolation, the spectral analysis shows a less pronounced obliquity peak than in the case of the initial fitting criteria. This is due to the fact that the model output do not match well the data on the [2000 - 2500] kyr BP period, where the optimal $V_0$ threshold is high in comparison to all other forcings.

This shows us that the correlation coefficient might not be the best criteria to determine which model output best fits our data. In this case (for the summer solstice insolation forcing on the [2000 - 2500] kyr BP period), the highest correlation coefficient is obtained for a relatively high $V_0$ threshold value ($V_0$ = 4.6), that leads to too large ice volumes compared to the data on this period.

[Figure]

Figure Q4-5a : *[Figure 5 of the manuscript]*

Normalized model results over the last million year, with the four different summer insolation forcings at 65° N when the accuracy criteria is the initial criteria of the manuscript.

[Figure]

Figure Q4-5b : *[Figure 5 of the manuscript with the correlation coefficient as fitting criteria]*

Normalized model results over the last million year, with the four different summer insolation forcings at 65° N when the accuracy criteria is the correlation coefficient.

On Figures Q4-5, the best fit over the last Myr period is displayed. Figure Q4-5a corresponds to the original figure of the manuscript, where the initial accuracy criteria is used, whereas Figure Q4-5b corresponds to the case where the correlation coefficient is used as fitting criteria.

On the [0 - 1000] kyr BP period, the optimal $V_0$ values obtained with the correlation coefficient as a fitting criteria lead to a greater model-data mismatch than in the case of our initial accuracy criteria. The deglaciations are not all at the right place.

To summarize, the exact values obtained with the coefficient correlation for the $V_0$ threshold and the corresponding model realizations are different from the values obtained when our initial accuracy criteria is used. However, this does not change the main results of the paper, as we are able to produce a shift from 41 kyr to larger cycles, by increasing the $V_0$ threshold, and thus for all insolation forcings.
Our criteria is better suited for this model (that is threshold - based) and the choice of the solution corresponding to the highest initial criteria gives better results than when choosing the one maximizing the correlation coefficient, in terms of deglaciations placement.

We suggest to add a reference to the answer to the reviewer when we describe the choice of the accuracy criteria l.154.

5. I am interested in knowing more about the optimisation procedure. For each period and insolation forcing, did you any optimisation algorithm or trial and error?

For the optimisation procedure for each period and insolation forcing, we used an optimisation algorithm, based on a trial and error method in an automated manner.

For each insolation forcing and period we have performed several simulations, with different $V_0$ and  different initial conditions. For each of these simulations, we computed our accuracy criteria and selected the simulation that gave the highest accuracy criteria.

In more details, the $V_0$ values considered are between 1 and 8 with a step of 0.1.  The initial condition corresponds to the couple ($V_{init}$, init_state) with $V_{init}$ the initial volume and init_state the initial state of the simulation (glaciation or deglaciation). The initial volume is taken between 0 and 5, with steps of 0.2. For every ($V_0$, $V_{init}$, init_state) possible triplet, we perform a simulation. This corresponds to 3500 simulations for each time period and insolation forcing. Then, we select the simulation with the highest accuracy criteria. This gives us the 'best $V_0$' value for a given period and insolation forcing. Here, we should stress that the use of different initial conditions ($V_{init}$, init_state) is done in order to compensate for the fact that we do not know the initial conditions at each time. Over periods of time containing relatively few cycles (like our 500 kyr periods), to start the model with initial conditions far from the data (for example with an high initial ice volume when the data at this time show a low volume and in the wrong state) would lead to a low accuracy coefficient as the first cycle will be 'misplaced', influenced by the initial conditions.

We propose to rephrase this part in the manuscript, and replace l.170- 171 by :

"The $V_0$ values that maximize the accuracy criteria for each time period and insolation forcing are called 'optimal $V_0$'. To determine the optimal $V_0$ threshold corresponding to each period and insolation forcing, several simulations were carried out and the parameter values maximizing the accuracy criteria $c$ were selected. More precisely, for each insolation and period, 3500 simulations corresponding to different $V_0$ thresholds (from $V_0$=1.0 to $V_0$=8.0 with a step of 0.1) and different initial conditions (initial volume $V_{init}$ ranging from 0.0 to 5.0 with a step of 0.2, and initial state - glaciation or deglaciation) were performed."

6. How do you define the "Best fit over the Quaternary"? (Section 3.2) This is not explained. I assume the authors use a V0 that changes with time (5 different values, in the 5 different periods considered), is that correct? Please, make explicit.

The reviewer is right in the interpretation of the "best fit over the Quaternary", and we agree that it is not clearly stated in the manuscript.

We suggest to add explanatory sentences at the end of section 2.3 (after the modification proposed above for l.170-171). In section 3.2, we will refer the reader to the definition given in section 2.3.

"For each insolation forcing, the best fit over the Quaternary is defined as the simulation over the whole Quaternary (0- 2500 kyr BP) with a $V_0$ changing with time, and that is equal to the corresponding optimal $V_0$ at each time period."

7/ For the "Best fit over the Quaternary" the authors use a V0 that changes with time. As a benchmark, I think the authors should also show how the performance with a changing V0 compare with the one of a constant V0. Please, repeat the optimisation procedure using the whole 0-2500 kyr BP period and provide the optimised constant V0 and corresponding model output. Please add these curves for each forcing in Figure 4. Also add in Figs. 1 and 2 the corresponding optimal constant-over-time V0 and accuracy when considering the entire 0-2500 kyr period.

In the next paragraphs, we call $V_0^Q$ the optimal $V_0$ threshold that is obtained when the optimization procedure is carried out over the whole Quaternary. The results in the case of a changing $V_0$ (corresponding on each period to the optimal $V_0$ threshold obtained on that period) are compared to the results in the case of a constant $V_0$ over the whole Quaternary ($V_0^Q$).

We repeated the optimization procedure over the whole quaternary.

Figure Q7 - 2 compares the optimal $V_0$ obtained for the whole Quaternary ($V_0^Q$) to the optimal $V_0$ for each period.

[Figure]

Figure Q7-2 : Optimal deglaciation threshold $V_0$ over the five different periods for the four different summer insolation forcings at 65° N, as well as the optimal constant $V_0$ threshold obtained when the optimization procedure is done over the whole Quaternary ($V_0^Q$). When several values of the deglaciation threshold $V_0$ maximize the accuracy criteria, the mean value is plotted and the other possible values are represented with errorbars.

The $V_0^Q$ is between 3.4 and 4 for each insolation type. It is a value in the middle of the highest values that best fit the latter part of the record and the lowest values that best fit the earliest part of the record.

Figure Q7-3a represents the corresponding accuracy obtained on the whole Quaternary with a constant $V_0^Q$ value (values represented by diamonds), as well as the accuracy when the $V_0$ value is varied over each period (initial case of the manuscript, represented by full bars). To obtain this figure, we used the $V_0^Q$ value previously obtained corresponding to each forcing, and performed simulations with this value over each 500 kyr period, with several initial conditions. The highest value of the accuracy criteria obtained was selected and corresponds to the diamonds values.

As expected, the accuracy with a fixed $V_0^Q$ value (diamonds) on each period and for each forcing is lower than or equal to the case when $V_0$ is being optimized (full bars). It is the case by definition, as the 'optimal $V_0$' were choosen maximizing the accuracy criteria. Indeed, the $V_0^Q$ value being in the middle of highest value that best fit the latest part of the record and lowest values that best fit the earliest part of the record, the model outputs with a constant $V_0^Q$ value have globally a poorer fit on all the record.

[Figure]

*Figure Q7-3a: Accuracy over the five different periods for the four different summer insolation forcings at 65° N, as well as the corresponding accuracy criteria over the Quaternary when the optimization procedure is done over the whole Quaternary. The diamonds correspond to the accuracy obtained on each period when taking the optimized constant $V_0$ on those periods instead of the optimal $V_0$.*

[Figure]

*Figure Q7-4: Best model fit over the whole Quaternary and corresponding spectral analysis when $V_0$ is varied over time (initial case of the manuscript displayed in full lines) and when $V_0$ is taken equal to $V_0^Q$ (dashed line).*

Figure Q7-4 displays the results over the whole Quaternary in the case where the $V_0$ value is changed over each period (full lines) and when it is kept to the constant value optimized over the whole Quaternary, $V_0^Q$ (dashed lines)

In the case of constant $V_0^Q$ value, the ice volume values obtained prior to the MPT are too high in comparison to the record. This is not surprising, as the $V_0^Q$ values for each insolation are globally higher than the optimal $V_0$ values over the pre-MPT period, leading to larger ice volume (as the deglaciation threshold is higher, larger volumes can form before the deglaciation takes place). On the contrary, on the latest part of the record, the model tends to deglaciate too often when using the fixed $V_0^Q$ values. This is also not surprising as the $V_0^Q$ values for each insolation are globally lower than the optimal $V_0$ values over the post-MPT period, allowing deglaciations to happen more often, when a lower ice volume is reached.

Figure Q7-5 displays the results over the last million year period, in the case where the $V_0$ value is set to its optimal value over the [0 - 1000 kyr] period (initial case of the manuscript, represented by full lines) and when it is set to the constant value optimized over the whole Quaternary, $V_0^Q$ (dashed lines). For all insolation forcings except the ISI above 300 W/m2, the fit to the data is better in the case where the $V_0$ value is set to its optimal value over the [0 - 1000] kyr BP period and not the whole Quaternary. In the case of the ISI above 300 W/m2 forcing, the results are identical as the optimal V0 value over the [0 - 1000] kyr BP period is equal to the $V_0^Q$ value.

[Figure]

Figure Q7-5 : Normalized model results over the last million year, with the four different summer insolation forcings at 65° N when the accuracy criteria is the initial criteria of the manuscript (full line) and when the $V_0$ value is taken equal to $V_0^Q$ (dashed lines).

We thank the reviewer for his/ her helpful suggestion to compare the results obtained to a fixed $V_0$ value over the whole Quaternary ($V_0^Q$). In the following, we propose additions to the manuscript to take into account this suggestion.

Concerning the figures, we prefer to keep Figure 4 and Figure 5 as in the original version of the manuscript, in order not to overload them.

However, we suggest to modify Figure 2 and Figure 3 of the manuscript. We suggest to replace Figure 2 by Figure Q7-2, to display the $V_0^Q$ values. We suggest to replace Figure 3 by Figure Q7-3b. In comparison to Figure Q7-3a, the diamonds values are not displayed anymore in Figure Q7-3b. We think that displaying the accuracy values for both cases (varied $V_0$ and fixed $V_0^Q$ ) would overload the figure, and we therefore prefer to add to the manuscript only the

accuracy value over the whole Quaternary when $V_0$ is being optimized over the whole Quaternary ($V_0^Q$).

[Figure]

*Figure Q7-3b: Accuracy over the five different periods for the four different summer insolation forcings at 65° N, as well as the corresponding accuracy criteria over the Quaternary when the optimization procedure is done over the whole Quaternary.*

We suggest the following additions to the text of the manuscript, following the suggested additions in question 5 and 6 of the reviewer (in square brackets in the following) :

"[The $V_0$ values that maximize the accuracy criteria for each time period and insolation forcing are called 'optimal $V_0$'. To determine the optimal $V_0$ threshold corresponding to each period and insolation forcing, several simulations were carried out and the parameter values maximizing the accuracy criteria *c* were selected. More precisely, for each insolation and period, 3500 simulations corresponding to different $V_0$ thresholds (from $V_0$=1.0 to $V_0$=8.0 with a step of 0.1) and different initial conditions (initial volume $V_{init}$ ranging from 0.0 to 5.0 with a step of 0.2, and initial state - glaciation or deglaciation) were performed. For each insolation forcing, the best fit over the Quaternary is defined as the simulation over the whole Quaternary (0- 2500 kyr BP) with a $V_0$ changing with time, and that is equal to the corresponding optimal $V_0$ at each time period]. Additionally, simulations were performed to determine the optimal $V_0$ threshold that is obtained when the optimization procedure is carried out over the whole Quaternary. It is called $V_0^Q$ in the following."

Additionally, we rephrase l.186 :

"For each insolation, the deglaciation threshold $V_0$ maximizing the accuracy for each of the five 500 kyr periods, as well as the fixed $V_0^Q$ value maximizing the accuracy over the whole Quaternary were computed. The results are displayed in Fig. 2."

We add at the end of l.194 :

"The optimal $V_0$ over the whole Quaternary, $V_0^Q$ is between 3.4 and 4 for each insolation type. It is a value in the middle of the highest values that best fit the latter part of the record and the lowest values that best fit the earliest part of the record."

We rephrase l.194 :

For each insolation, the accuracy corresponding to the optimal $V_0$ threshold for each time period as well as to the fixed $V_0^Q$ value maximizing the accuracy over the whole Quaternary is displayed in Fig. 3.

We add at the end of l.198 :

"The accuracy obtained on the whole Quaternary period (fixed $V_0^Q$ value) is globally lower than the accuracy on each time period. This is due to the fact that the $V_0^Q$ values obtained are lower than the optimal $V_0$ values on the later part of the Quaternary and higher than the optimal $V_0$ values on the earliest part of the Quaternary, leading to a poorer representation of both of these periods"

In the following tables, values of the experiments are displayed.

| | Optimal $V_0$ over the [0 - 1000] period | whole Quaternary $V_0^Q$ value and corresponding accuracy over the [0 - 1000 ] period |
|---|---|---|
| summer solstice | 5.1 | 4.0 |
| caloric season | 4.65 | 3.5 |
| ISI > 300 W/m2 | 3.9 | 3.9 |
| ISI > 400 W/m2 | 4.75 | 3.4 |

| | Corresponding accuracy to the optimal V0 over the [0 - 1000] period | corresponding accuracy to the whole Quaternary V0Q value over the [0 - 1000 ] period |
|---|---|---|
| summer solstice | 0.92 | 0.73 |
| caloric season | 0.82 | 0.48 |
| ISI > 300 W/m2 | 0.87 | 0.87 |
| ISI > 400 W/m2 | 0.82 | 0.48 |

Table : optimal $V_0$ threshold over each period

| | summer solstice | caloric season | ISI above 300 W/m2 | ISI above 400 W/m2 |
|---|---|---|---|---|
| | | | | |
| [0 - 1000] kyr BP | 5.1 | 4.65 | 3.9 | 4.75 |
| [0 - 500] kyr BP | 5.25 | 5.9 | 3.85 | 4.75 |
| [500 - 1000] kyr BP | 4.85 | 4.4 | 4.5 | 4.5 |
| [1000 - 1500] kyr BP | 3.05 | 3.05 | 4.1 | 3.2 |
| [1500 - 2000] kyr BP | 3.85 | 1.7 | 2.0 | 1.95 |
| [2000 - 2500] kyr BP | 3.3 | 2.7 | 2.7 | 2.9 |
| [0 - 2500] kyr BP (whole Quaternary) | 4.0 | 3.5 | 3.9 | 3.4 |

Table : highest accuracy obtained over each period and accuracy on each period corresponding to the $V_0^Q$ value (in parenthesis)

| | summer solstice | caloric season | ISI above 300 W/m2 | ISI above 400 W/m2 |
|---|---|---|---|---|
| [0 - 1000] kyr BP | 0.92 (0.73) | 0.82 (0.48) | 0.87 (0.87) | 0.82 (0.48) |
| [0 - 500] kyr BP | 1.0 (0.64) | 0.83 (0.64) | 0.77 (0.77) | 0.77 (0.46) |
| [500 - 1000] kyr BP | 0.86 (0.81) | 0.86 (0.59) | 1.0 (0.8) | 0.86 (0.76) |
| [1000 - 1500] kyr BP | 0.73 (0.49) | 0.64 (0.64) | 0.42 (0.25 | 0.73 (0.73 |
| [1500 - 2000] kyr BP | 0.55 (0.48) | 0.78 (0.46) | 0.56 (0.36) | 0.82 (0.3) |
| [2000 - 2500] kyr BP | 0.45 (0.36) | 0.68 (0.45) | 0.76 (0.45) | 0.61 (0.55) |
| [0 - 2500] kyr BP (whole Quaternary) | 0.51 (0.49) | 0.65 (0.55) | 0.60 (0.49) | 0.70 (0.54) |

**Specific comments**

L 25-26: "…due to reduced summer insolation, at latitudes typical of Northern Hemisphere ice sheets, 65° N." I suggest to change for "…due to reduced summer insolation, at latitudes of the Northern Hemisphere critical for ice sheet growth (65°N)."

We accept the suggested change.

L 38: independent --> independent
This will be modified in the next version of the manuscript.

L 49: the more insolation --> the highest insolation
This will be modified in the next version of the manuscript.

L 53: more adapted --> better
This will be modified in the next version of the manuscript.

L 71: explicitely --> explicitly
This will be modified in the next version of the manuscript.

L 77-77: In Equation (1) the labels (g) and (d) are misplaced
This will be modified in the next version of the manuscript.

L 80: "…a typical latitude for Northern Hemisphere ice sheets." This part of the sentence has already been used in the introduction, no need to repeat it.
We will remove this part of the sentence in the next version of the manuscript.

L 93-93: "The importance of orbital forcing alone seems able to start a glaciation" This sentence does not make sentence, please reformulate.
We propose to rephrase the sentence and to invert the order of the paragraph for clarity. This gives :
"A critical point is to define the criteria for the switch between the glaciation and deglaciation states. To enter the deglaciation state, both ice volume and insolation seem to play a role (Raymo, 1997; Parrenin and Paillard, 2003, 2012), as terminations occur after considerable build-up of ice sheet over the last million year. To represent the role of both ice volume and insolation in the triggering of deglaciations, the condition to switch from (g) to (d) state uses a linear combination of ice volume and insolation. The deglaciation is triggered when the combination crosses a defined threshold $V_0$ : the deglaciation threshold. As in the work of Parrenin and Paillard (2003), this allows transitions to occur with moderate insolation when the ice volume is large enough and reciprocally. On the contrary, glacial inceptions seem to depend on orbital forcing alone (Khodri et al, 2001; Ganopolski and Calov, 2011). Therefore, the condition to switch from the deglaciation state to the glaciation state is based on insolation only : it is possible to enter deglaciation when the insolation becomes low enough."

L 169: "In order to study the evolution of the optimal model parameters over the Quaternary…" Please correct to indicate that only V0 is being optimised.
We suggest to replace the sentence by : "In order to study the evolution of the optimal deglaciation threshold $V_0$ over the Quaternary, it was divided into five 500 kyr periods."

L 170: periods --> period
This will be modified in the next version of the manuscript.

L 173: "…the best fit parameters…" please modify to indicate only V0 is being fit
We suggest to replace the sentence by : "To calculate our accuracy criteria c and therefore determine the optimal $V_0$ threshold over a given period, a definition of the deglaciation in the data is needed."

L 183-184: "The model state was compared to the middle of the deglaciation." I don't understand this sentence

We suggest to rephrase in this way :
"To determine if a deglaciation is well reproduced by our model, we look at the state of the model (glaciation or deglaciation) at the time halfway between the start of the deglaciation and the end of the deglaciation. If the model state at that time is "deglaciation", the deglaciation is considered as correctly reproduced."

L 233: varying --> varying
This will be modified in the next version of the manuscript.

L 242: varies --> varies
This will be modified in the next version of the manuscript.

L 273-274: "To model future natural evolutions of the climate system, one would need to take into account for possible evolutions of the V0 threshold." Please acknowledge that, possibly, the other parameters might as well change.

Indeed. We suggest to add an additional sentence l.274 :
"To model future natural evolutions of the climate system, possible evolutions of the $V_0$ threshold should be considered. However, we do not exclude the fact that variations of other parameters, that were kept constant in this study, could vary in the future. For instance, different $I_0$ thresholds have to be considered."

L 283: Also add here Talento and Ganopolski, 2021 as reference.
This will be done in the revised manuscript.

L 287: "…very few tunable parameters… " in fact, you should clarify that you tune only 1 parameter.

We suggest adding the following sentence at the start of l.288. "Only one parameter was varied, the deglaciation threshold parameter V0, while the other were kept constant."

Section 2.4: I think a plot of the d18O record used with indications of the start and end of deglaciations, according to your criterion, is due here. Or at least refer the reader to this information in figure 3.

We would prefer not to add a plot here, in order to keep the figures for the core message of the paper. However, we agree that we should more clearly refer the reader to this information in the existing figures Figure 4 and Figure 5. We suggest to add a sentence at the end of line 182 : "the deglaciation periods in the data corresponding to the time between the deglaciation starts and deglaciation ends are displayed with a blue shading in Fig. 4 and Fig. 5".
In the legend of Fig. 4 and Fig 5. the label of the blue and yellow colors have been swapped. The blue color corresponds to the deglaciation in the data, and the yellow color to the deglaciation state in the model. This will be corrected in the next version.

Figure 1: Normalized to what? by standard deviation? Please clarify. The ylabel in blue line must be corrected.

The insolation plots are centered and normalized to their standard deviation. For the spectral analysis curves in the new figure proposed, the spectral power is normalized by the standard deviation. The spectral analysis is performed using insolation curves over the [0 - 2500] kyr BP. This will be clarified in the legend of Fig 1.

Why show only until 1000 kyr BP when the totality of the Quaternary (0-2500 kyr BP) is the focus on the rest of the paper? Please show the plot considering 0-2500 kyr BP.

While the spectral analysis was carried out on insolation on the totality of the Quaternary ([0 - 2500] kyr BP), the insolation was initially only plotted on the [0 - 1000] kyr BP period, to make it easier for the reader to see the differences between the differents curves. However, we agree that this might be confusing as the paper focuses on the whole Quaternary period and suggest to modify Figure 1 accordingly.
The new figure is displayed below.

[Figure]

*Figure : [New version of Fig. 1 of the manuscript ]*

*(a) The four different summer insolation types at 65° N (centered and normalized by standard deviation). (b) Corresponding spectral analysis, normalized by the standard deviation.*

In all the text: The acronym ISI was introduced in line 5, but frequently not used afterwards.
Indeed, we will use it more frequently after its definition in the revised manuscript.

When starting a new paragraph, sometimes there is an indent, sometime there is not.
We will modify it in the revised manuscript.